# Dissecting the sequence determinants for dephosphorylation by the catalytic subunits of phosphatases PP1 and PP2A

Bernhard Hoermann [1,2,3,4], Thomas Kokot [1,2], Dominic Helm [5], Stephanie Heinzlmeir [6,7], Jeremy E. Chojnacki [1,2,3], Thomas Schubert [2,8], Christina Ludwig [7], Anna Berteotti [3], Nils Kurzawa [3,4], Bernhard Kuster [6,7], Mikhail M. Savitski [3,5] & Maja Köhn [1,2,3 ✉]

The phosphatases PP1 and PP2A are responsible for the majority of dephosphorylation reactions on phosphoserine (pSer) and phosphothreonine (pThr), and are involved in virtually all cellular processes and numerous diseases. The catalytic subunits exist in cells in form of holoenzymes, which impart substrate specificity. The contribution of the catalytic subunits to the recognition of substrates is unclear. By developing a phosphopeptide library approach and a phosphoproteomic assay, we demonstrate that the specificity of PP1 and PP2A holoenzymes towards pThr and of PP1 for basic motifs adjacent to the phosphorylation site are due to intrinsic properties of the catalytic subunits. Thus, we dissect this amino acid specificity of the catalytic subunits from the contribution of regulatory proteins. Furthermore, our approach enables discovering a role for PP1 as regulator of the GRB-associated-binding protein 2 (GAB2)/14-3-3 complex. Beyond this, we expect that this approach is broadly applicable to detect enzyme-substrate recognition preferences.

[1] Faculty of Biology, Institute of Biology III, University of Freiburg, Freiburg, Germany. [2] Signalling Research Centres BIOSS and CIBSS, University of Freiburg, Freiburg, Germany. [3] European Molecular Biology Laboratory, Genome Biology Unit, Heidelberg, Germany. [4] Collaboration for joint PhD degree between EMBL and Heidelberg University, Faculty of Biosciences, Heidelberg, Germany. [5] European Molecular Biology Laboratory, Proteomics Core Facility, Heidelberg, Germany. [6] Chair of Proteomics and Bioanalytics, Technical University of Munich (TUM), Freising, Germany. [7] Bavarian Center for Biomolecular Mass Spectrometry (BayBioMS), Technical University of Munich (TUM), Freising, Germany. [8] Signalling Factory, University of Freiburg, Freiburg, Germany. ✉email: maja.koehn@bioss.uni-freiburg.de

Phosphorylation on Ser and Thr residues is among the most common post-translational modification (PTM) in mammals and as such plays significant roles in the regulation of cellular processes[1]. Aberrations in phosphorylation patterns are tightly linked to the initiation and progression of a multitude of diseases[2–8]. Phosphorylation of proteins is mediated by kinases and counteracted by phosphatases, which hydrolyze phosphomonoesters. More than 400 genes encoding Ser/Thr-specific human kinases are counterbalanced by about 40 genes encoding protein phosphatases (PPs)[8]. However, the Ser/Thr-specific phosphoprotein phosphatase family (PPP) achieves a similar complexity like that of kinases by existing as holoenzymes in vivo with many different regulatory subunits. Of those PPPs, PP1 and PP2A are the two major phosphatases and collectively have a large number of substrates that have been difficult to assign to one PPP or the other[6,9,10].

In contrast to kinases, substrate specification of PP1 and PP2A is a multi-layer system: holoenzyme formation of the catalytic subunit (PP1c/PP2Ac) bound to one or two out of many regulatory subunits determines the substrate specificity[6,7]. These holoenzymes recruit the substrates before the dephosphorylation is carried out by the catalytic core. Nevertheless, surrounding the active site, PP1c and PP2Ac have three grooves with distinct properties, the acidic, the hydrophobic, and the C-terminal groove[11,12]. Another, more basic layer for substrate recognition than holoenzyme formation mediated by these grooves has been suggested[13] but is still under debate[7,14] and so far could not be studied independently from holoenzyme formation on the proteome level due to methodical limitations. Despite high conservation between PPPs, differences in the active site composition and the three grooves have been shown to be the underlying causes for different potencies of natural, small molecule inhibitors such as microcystin[12,15,16], okadaic acid[17], and tautomycin/tautomycetin[18,19]. Furthermore, early studies using small sets of synthetic phosphopeptides already suggested a preference of PP1 and PP2A for pThr over pSer[20–22], as well as for basic over acidic residues N-terminal[22,23] and a disfavoring of Pro in position +1 relative to pThr/pSer[22]. More recent findings at the holoenzyme level were able to underline the biological relevance in mitosis for the selectivity of PP2Ac in complex with its B55 subunit for pThr[24–27], concluding conversely on the one hand that B55 was responsible for the pThr selectivity[27] and on the other hand that it was due to the intrinsic preference of PP2Ac[24]. In addition, basic substrate motifs and pThr were discovered to be preferentially dephosphorylated during mitotic exit, but without dissecting the preferences of PP1 and PP2A towards these motifs[25]. Also, analysis of the residues surrounding the phosphorylation (p)-sites of the less than 80 established p-sites in protein substrates for each phosphatase, which is a low number compared to possible substrates and could have introduced a bias for amino acid sequences, showed the occurrence of Arg at the N-terminus close to the p-site for PP1 substrates, but not for PP2A[8]. However, none of these studies could provide direct, unbiased evidence for intrinsic selectivity of the catalytic subunits due to the holoenzyme-based setups, resulting in different interpretations such as the intrinsic PP2Ac versus B55-directed specificity for pThr[24,27]. In case synthetic peptide libraries were used to study phosphatase substrate specificity, either dephosphorylation kinetic analysis was done for single peptides[20–23], or microarrays were employed[28–30]. While offering higher throughput than single peptide analysis, detection of dephosphorylation on the microarray requires a negative read-out of loss of binding of phospho-specific antibodies, severely limiting the sensitivity and reliability of these assays[28–30]. The result of these attempts to study the specificity of the catalytic subunits PP1c and PP2Ac is that they are currently assumed to have little appreciable substrate specificity[7,9,13]. Mass spectrometry (MS) read-outs of the dephosphorylation of phosphopeptide libraries would enable a direct read-out and higher throughput than microarray technology. However, so far randomized synthetic peptide libraries have rather been used for technology-oriented MS applications, for example for search engine optimization, investigation of fragmentation patterns and retention time predictions[31], or for binding assays where the proteins that bound to the immobilized libraries were identified[32]. Therefore, studies disentangling the two layers for substrate recognition, with the catalytic subunit recognizing motifs around the p-site and regulatory proteins binding at interfaces more distant from the p-site, are still lacking, but are required to get a comprehensive insight into PP1 and PP2A regulation.

In this study, we overcome the limitations of current experimental approaches by developing a proteomic strategy employing randomized synthetic phosphopeptide libraries that are much larger than those applied previously. We apply these libraries to study the intrinsic substrate preference of PP1c and PP2Ac using MS as read-out, allowing for a direct detection of dephosphorylation. We complement this non-natural in vitro approach by a phosphoproteomic approach on protein substrates that we develop to enable reducing the occurrence of indirect dephosphorylation events, in order to compare the specificity toward thousands of phosphopeptides from the library to protein substrates on a proteome-wide scale. Our results reveal comprehensive, unbiased insights into the contribution of the catalytic subunits to PP1 and PP2A selectivity, and deliver a plethora of substrate candidates.

## Results

**Design, synthesis, MS-validation of phosphopeptide libraries.** In order to determine phosphatase specificity on the phosphopeptide level, we developed an in vitro phosphopeptide library dephosphorylation followed by mass spectrometry approach (PLDMS, Fig. 1a; see the methods for details). In this approach, phosphopeptide libraries would be treated with PP1, PP2A, or left untreated, purified, and analyzed by LC-MS/MS measurements. In the first step, design, synthesis, and validation of phosphopeptide libraries were required. The phosphopeptide libraries were designed to yield an even distribution of amino acids in the different randomized positions and to give high coupling efficiency during synthesis, thus reducing the number of by-products. This, in turn, would help to ensure high MS data measurement quality by giving multiple coverage of the peptide masses. To fulfill these requirements, we used 10-mer peptides with 14 different amino acids and pSer/pThr as the fifth amino acid (position 0, Fig. 1b) and designed five different libraries (four N-terminal: Nterm, and one C-terminal: Cterm, Fig. 1b) based on the following considerations.

To limit the complexity of the library and increase data quality during analysis, we anticipated 5000–6000 theoretical peptides per library as an optimal complexity, since this would result in 10,000-12,000 theoretical masses for peptides upon dephosphorylation, which would still lead to redundant measurements of the same peptide during LC-MS/MS runs. To reduce the complexity by rational design, known PP1 and PP2A protein substrates were analyzed (see the methods). This revealed that the most influence on PP1 substrate recognition is within positions −4 to +3 relative to pSer/pThr. For PP2A, no preference was found[8]. The Nterm libraries were randomized at three of the relevant four positions N-terminal of pSer/pThr, and an Ala was placed at the non-randomized position as commonly done in alanine scans[33]. Ala was also used as neutral placeholder at the C-terminus, as commonly done in inverse alanine scans[34]. Lys was added at the

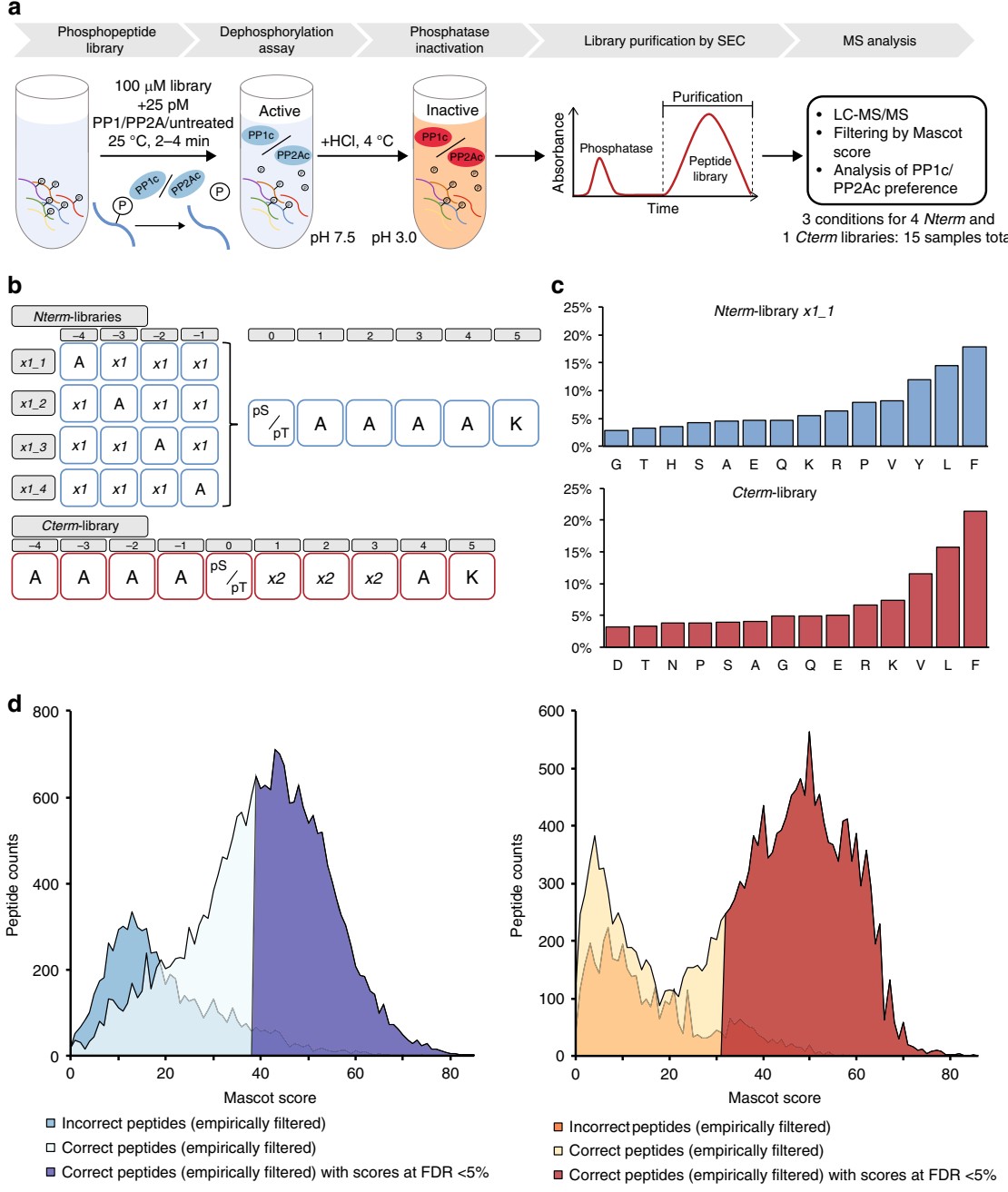

**Fig. 1 Design of peptide libraries and validation by LC-MS/MS. a** Scheme of the experimental PLDMS workflow. **b** Phosphopeptide libraries used for substrate preference evaluation. **c** Amino acid distribution in the permutated positions (*x1* and *x2*) of the *Nterm_x1_1* (blue) and the *Cterm* library (red) after reference measurement (samples not treated with PP1 or PP2A, i.e. untreated) and filtering for expected sequences (see Supplementary Table 1 and the methods section). Perfectly random incorporation of all amino acids would result in 7% per amino acid. Source data are provided as a Source Data file. **d** Mascot Score (statistical value for how well detected data matches database sequences) distributions of the reference measurements for *Nterm_x1_1* (left) and *Cterm* library (right). Empirically wrong peptides are peptides with sequences not matching expected sequences from the synthetic route. For the reference measurement, empirical filtering by Mascot Score cut-offs of 39 and 32 at a false discovery rate (FDR) of 5% for the *Nterm_x1_1* and the *Cterm* library, respectively, allowed separation of correct peptides from wrong ones.

*C*-terminus to ensure the detectability of the peptides by MS. The *Cterm* library was designed accordingly. The design resulted in 25,196 different theoretical phosphopeptides.

For random incorporation of amino acids during peptide synthesis, we optimized amino acid mixtures to yield equimolar peptide products[35] and assessed the amino acid distribution by LC-MS/MS analysis (Fig. 1c, Supplementary Tables 1, 2). The controlled randomization then allowed filtering out incorrect peptides according to expected peptides and including a 5% false

discovery rate (FDR, Fig. 1d). Finally, we obtained a library coverage of theoretical masses of 42% (2319 unique peptide sequences) for *Nterm* library *x1_1* and 31% (1684 unique peptide sequences) for the *Cterm* library. These datasets were used as the reference measurements for the following experiments.

**PP1c and PP2Ac substrate specificities for phosphopeptides**. The individual libraries were incubated with recombinant PP1c or

PP2Ac (Supplementary Fig. 1a) and the reaction was stopped by shifting the pH from 7.5 to 3 when 30–50% of the library was dephosphorylated, depending on phosphatase activity (see the methods for details, Supplementary Fig. 1b). Peptides were then separated from the recombinant protein by size-exclusion chromatography (SEC) (Supplementary Fig. 1c) and analyzed by LC-MS/MS (Fig. 1a). Analysis revealed that more than 75% of all data points could be attributed to correct peptides (Supplementary Table 3). Interestingly, of all erroneous data points, prediction of phosphorylation in positions that can be empirically excluded based on the synthesis, represented the largest portion. This demonstrated the power of empirical filtering to overcome the limitations of widely used false localization rates (FLRs) for MS-based phospho-motif studies. Further quality control included analysis of reproducibility, amino acid distribution between residues and *Nterm* libraries, as well as a potential impact of peptide phosphorylation on peptide (Mascot) scores (Supplementary Fig. 2a–d, Supplementary Table 3). After sample-specific FDR-based filtering (Supplementary Fig. 2e, Supplementary Data 1), 9072–9513 different sequences for the *Nterm* libraries and 2065–2362 different sequences for the *Cterm* library (ranging over the different treatments) entered further analysis (Supplementary Table 3).

We found that despite a distribution of 60% Ser to 40% Thr among all libraries (based on MS peptide counts), 73% of all sites dephosphorylated by PP1c and 83% for PP2Ac were pThr (Fig. 2a), demonstrating a global preference of both catalytic subunits for pThr over pSer, independent of the sequence context. The controlled setup then allowed calculating a normalized heat map indicating the preference of PP1c and PP2Ac for each amino acid independent of the occurrence of the amino acid in each position by comparing the rate of dephosphorylation of all peptides with a certain amino acid in a certain position to the average dephosphorylation rate of the library (see the methods). For PP1c we observed a preference for the positive charges of Lys and Arg over the negative charges of Glu in positions −4 to −1, with Lys being preferred at positions −4 and −1, but not at −3 (Fig. 2b). In positions +1 to +3, tendencies were the same for Lys/Arg preference over Asp/Glu, with Arg in +1 being the strongest preference. In contrast, PP2Ac showed a preference for negative charges instead of positive charges in positions −4 to −1. As PP1c, PP2Ac displayed a strong preference for Arg in position +1 (Fig. 2b). PP2Ac disfavored the naturally frequently occurring Pro in position +1, which reflects the S/TP motif recognized preferentially by some kinases and phosphatases[8], and PP1c showed a slight preference for Pro in position +1. This finding was rather surprising, given that PP1 and PP2A are the major Ser/Thr-specific phosphatases, which could have suggested an active-site-mediated recognition of the kinase S/TP motif to counterbalance kinase activity. Nevertheless, these results are largely in line with early studies based on small sets of peptides[22]. All these trends were also statistically significant when comparing the differences between PP1c-/PP2Ac in a third differential heat map (Fig. 2b). The coverage of several thousand peptides then allowed to analyze combinatorial effects: PP1c preferences in position −4 to −1 revealed that a second Arg decreased dephosphorylation efficiency and that the negative effects of Glu were additive (Fig. 2c). While PP1c displayed equal preference for one or two Lys, PP2Ac disfavored the presence of two Lys (Fig. 2c). Since all four *Nterm* libraries were synthesized and processed independently, we could demonstrate reproducibility of the observed tendencies (Supplementary Fig. 3a) and that peptides spanning the whole range of biophysical properties were included in our analysis (Supplementary Fig. 3b). To further validate these findings, we re-synthesized, purified, and quantified six phosphopeptide sequences and carried out

quantitative assays using PP1c and PP2Ac. Using the same basic sequence but with either pThr or pSer, a clear preference of both phosphatases for pThr was observed also in this non-competitive assay setup with no other peptides present, reflected by a higher catalytic efficiency ($k_{cat}/K_m$) (Fig. 2d). Also in agreement with the library results, the different preferences of PP1c toward Lys-, Glu-, and Pro-containing sequences were clearly reflected in this assay. In contrast to PP1c, PP2Ac slightly preferred the Glu sequence over the Lys-containing peptide but was unable to dephosphorylate a peptide containing Pro in +1 (Fig. 2e, Supplementary Fig. 3c). Interestingly, the general tendencies of PP1's preference for positive charges over PP2A are in line with the starting point of the library design based on *bona fide* substrate sites identified on the holoenzyme level[8].

**Substrate preference of PP1c and PP2Ac at the proteome level.** Next, we sought to validate these findings at the protein level in cell-based assays. In order to block dephosphorylation during cell lysis, we first inhibited the activity of endogenous PPPs in HeLa cells using the potent inhibitor Calyculin A (20 nM)[36]. We then lysed cells, thereby disrupting signaling pathways, and subjected three replicates of lysates to dephosphorylation assays using recombinant phosphatase (1 µM) followed by phosphoproteomic analysis (Fig. 3a). We identified a total of 3200 high confidence (class I) p-sites (Fig. 3b, Supplementary Fig. 4a, see methods for details). Of these class I p-sites, 75% were dephosphorylated upon phosphatase treatment. We found 1967 dephosphorylated by PP1 and 1840 by PP2A (Supplementary Data 2), with more than 50% of the p-sites showing different responses to PP1c and PP2Ac treatment (Fig. 3c), challenging the notion of these catalytic subunits having little appreciable intrinsic substrate specificity. Among all detected p-sites, irrespective of phosphatase treatment, more than 92% were found to be pSer. Accordingly, the majority of dephosphorylation happened on pSer. Importantly however, within the sets of dephosphorylated pSer and pThr sites the preference for pThr was confirmed for both phosphatases (Fig. 3d).

We then again sought to investigate the preference of the enzymes for amino acids surrounding the p-site in proteins, independently of the amino acid occurrence in a certain position. To this end, the relative abundance for each amino acid in a certain position was calculated in phosphatase-sensitive or insensitive groups. This rate of occurrence was then divided, resulting in fold-over-/under-representations (Fig. 4a). To highlight differences between phosphatases, these fold changes were then also compared between PP1c and PP2Ac (Fig. 4b). Of note, in these datasets about half of the detected amino acids at position +1 were Pro due to its natural occurrence in the S/TP motif. The counts for other amino acids were rather low and their significance for the analysis should be treated with caution. Therefore, we focused on the other positions in the analysis. For PP1c, the phosphoproteomic data agreed well with our findings from the peptide library with respect to the preference toward basic amino acids over acidic ones, particularly for Arg in position −3, and Lys in −1 (Fig. 4a). However, single positions differed from the library results, namely Lys at position −3 was not disfavored, while Lys at position +2 was favored more strongly in the phosphoproteomic setting, showing an even stronger preference for basic charges. The observed preferences for PP2Ac were also in broad agreement with the library results, especially the disfavor for Lys at the *N*-terminus, which differed significantly from PP1c. However, the preference for Glu at the *N*-terminus as well as the strong disfavor for Pro in +1 observed in the library was not recapitulated in the phosphoproteomic assay, and the presence of Arg at the *N*-terminus had a neutral

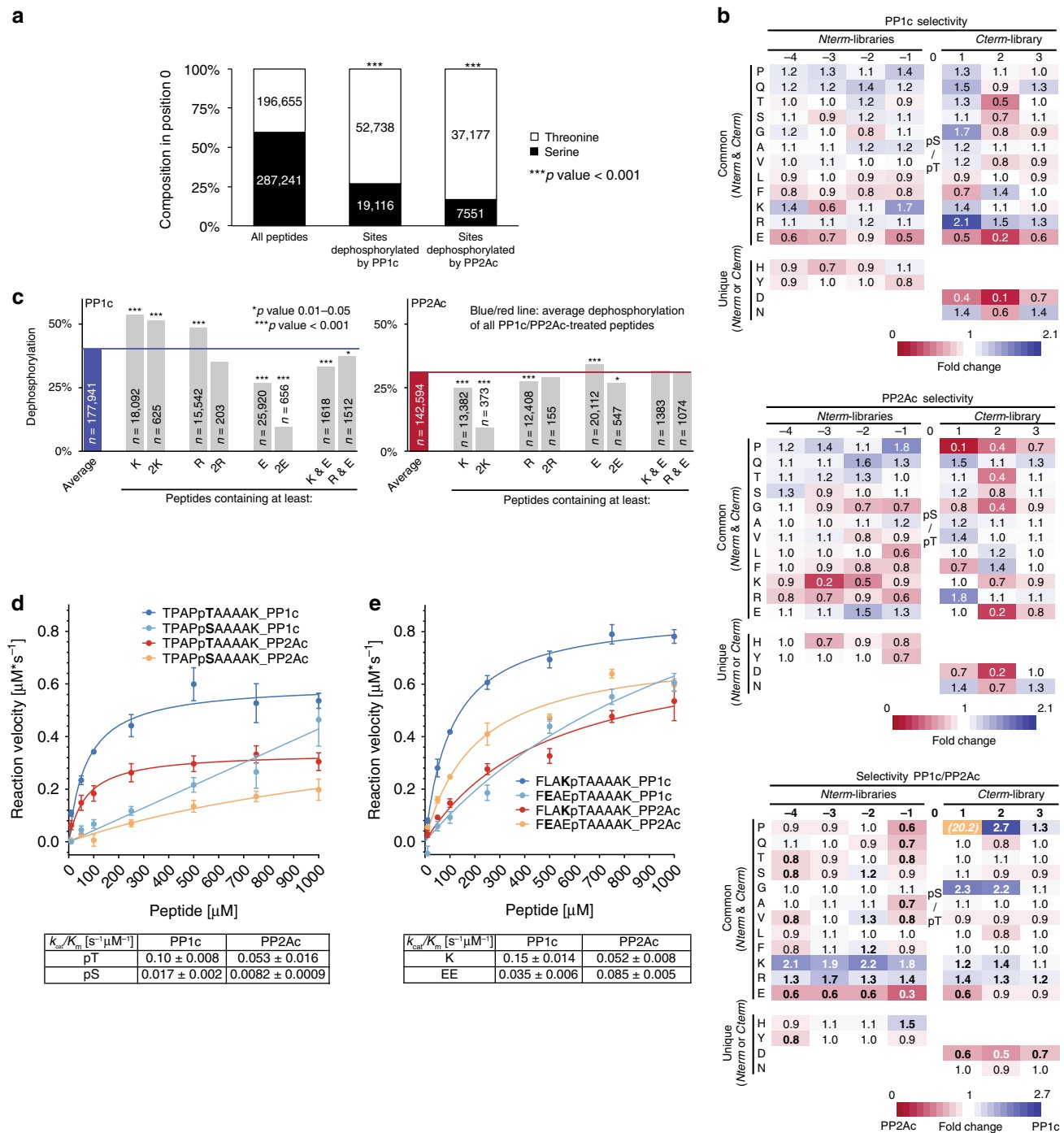

**Fig. 2 Analysis of PP1c/PP2Ac amino acid preference using the PLDMS approach. a** PP1c and PP2Ac preferably dephosphorylate pThr over pSer. For statistical analysis, a two-sided Fisher's exact test was used. **b** Heat-map analysis of the sequence context surrounding pSer/pThr dephosphorylated by PP1c and PP2Ac. Color coding: Phosphorylation-fold-change of all amino acids compared to the average library dephosphorylation. At 1 the dephosphorylation rate of an amino acid is equal to the average dephosphorylation rate. For a third heat map comparing PP1c and PP2Ac, dephosphorylated peptides for each amino acid were divided and *Nterm/Cterm* heat-maps were normalized by the median. To allow better visualization by color coding, Pro in+1 was excluded. Fold-changes >1.2 with an adjusted *p*-value <0.01 according to Fisher's exact test in the comparison of PP1c/PP2Ac are highlighted in bold. Please see the methods section and Source Data for details on statistics. **c** Analysis of the additive influence of positive and negative charges on dephosphorylation by PP1c/PP2Ac of *Nterm/Cterm* libraries. *p*-Values were derived by carrying out a two-sided Fisher's exact test. Exact *p*-values are provided in the Source Data. **d**, **e** Kinetic analysis of the dephosphorylation rate of four synthetic peptides upon PP2Ac or PP1c treatment. Error bars represent s.e.m. of three independent repeats with technical duplicates. $k_{cat}/K_m$ was calculated by comparison to a phosphate standard curve. Source data underlying Fig. 2a–e are provided as a Source Data file.

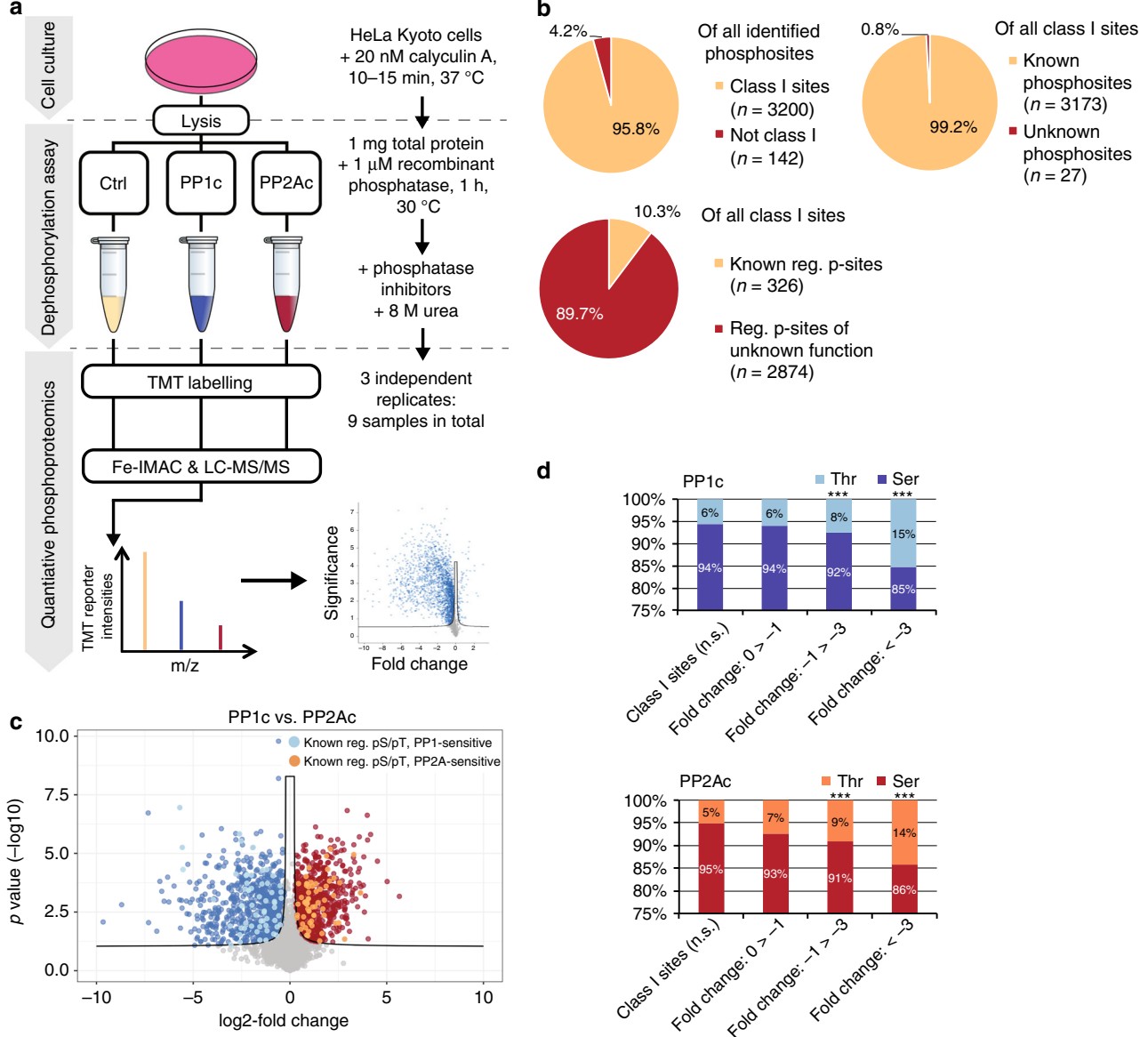

**Fig. 3 Phosphoproteomics confirm the substrate specificity of PP1c and PP2Ac. a** Workflow of phosphoproteomic experiments. **b** Identified p-sites were filtered for class I sites (localization probability of phosphorylation >0.75) and compared to known phosphosites from PhosphositePlus (v03/07/18) (Supplementary Data 2). **c** Differentially downregulated p-sites between PP1 and PP2A as determined by Student's *t* test (two-sided, FDR 0.01) (see Supplementary Fig. 4b). **d** PP1c and PP2Ac prefer pThr over pSer. The percentage of Thr found in PP1c/PP2Ac dephosphorylated p-sites with a log2 fold-change < −1 is significantly larger than the percentage of pThr in p-sites with a log2 fold-change > −1 according to a two-sided Fisher's exact test (***: *p*-value <0.001; 1.175e−06 (PP1), 2.426e−06 (PP2A)). n.s.: not significant. Source data of Fig. 3b, d are provided as a Source Data file.

effect (Fig. 4a). Differences for Pro in +1 can be explained by the previously observed dependency of PP2A for stabilization of Pro in *trans*[37], which cannot be achieved on the peptide level, and the high natural occurrence of Pro at this position could also increase the likelihood of dephosphorylation. Furthermore, the variances for Glu and Arg could reflect differences between the two set-ups. For instance, motif-based effects, which are important in intrinsically disordered regions and in phosphopeptides, could be overruled by structural effects based on the tertiary structure in ordered protein regions and are therefore less visible in a proteome-based setup. Another explanation could be that a partial holoenzyme formation of the recombinant phosphatases could take place in the lysate for phosphoproteomics, but not in the PLDMS approach. To control for possible holoenzyme formation, we carried out the assay presented in Fig. 3a and coupled it to gel-filtration analysis spanning molecular weights

(MW) of 30–700 kDa (Supplementary Fig. 5). The vast majority of PP1c/PP2Ac was found to be unassociated with higher MW complexes with only a very small fraction of PP1c potentially binding to regulators (Supplementary Figs. 6, 7). Therefore, significant signal coming from holoenzyme formation in the phosphoproteomic experiment is unlikely.

Taken together, the results from the PLDMS and the phosphoproteomic approaches confirm the intrinsic recognition of basic amino acids at the N-terminus by the catalytic subunit of PP1, which is the opposite for PP2A when comparing the amino acid specificity between both phosphatases (Figs. 2b, c; 4a, b). Also, both approaches confirm the preference for pThr of the catalytic subunit of both phosphatases (Figs. 2a, 3d). These results support a two-layered system for substrate specificity, with intrinsic preferences of PP1/PP2A catalytic subunits for p-site motifs on the one hand, and substrate specification by

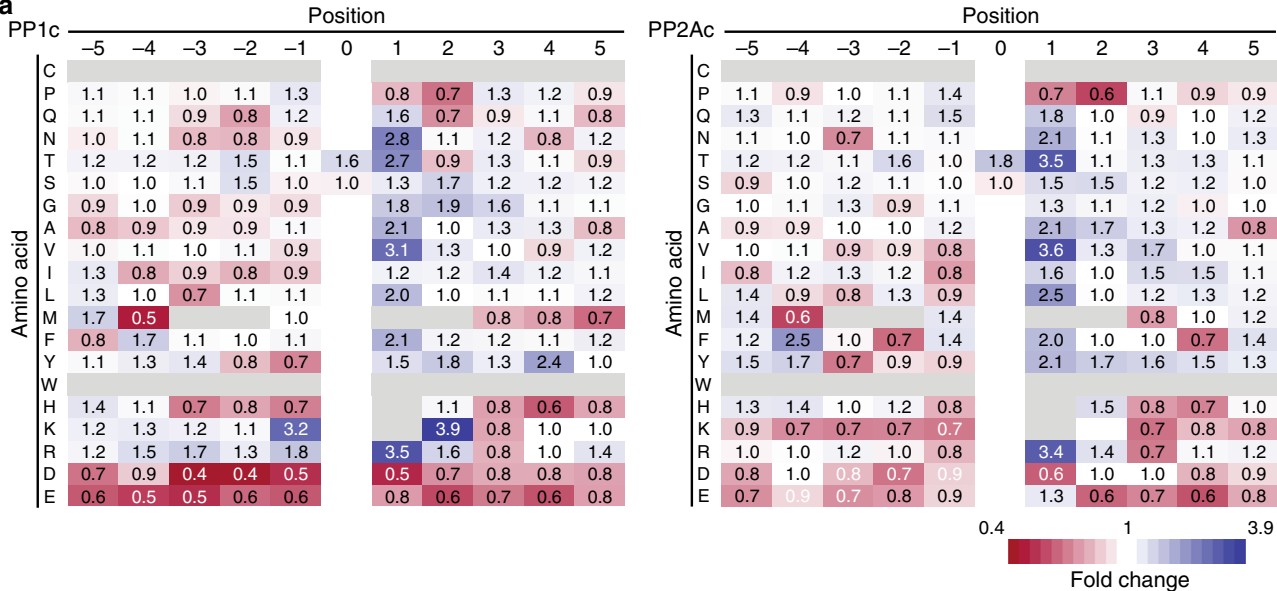

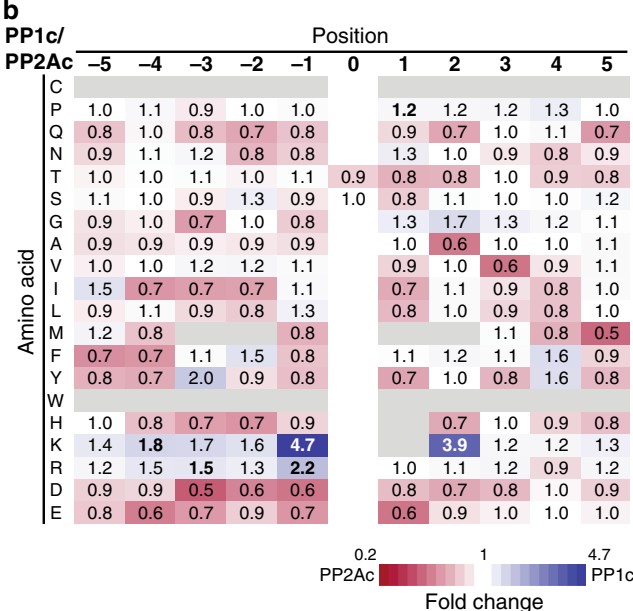

**Fig. 4 Heat-map analysis of PP1c/PP2Ac amino acid preference.** Amino acid preference surrounding pSer/pThr for PP1 and PP2A on the protein level from the phosphoproteomic data (Supplementary Data 2). Amino acids with <25 raw counts in a given position were excluded and are marked in gray. **a** Color coding represents fold-change of the relative abundance of a given amino acid in a given position between phosphatase-sensitive/insensitive phosphorylation sites. **b** Direct comparison of PP1c and PP2Ac fold changes displayed in (**a**). Blue highlights amino acids selectively preferred by PP1c, and residues statistically different between PP1c and PP2Ac (adjusted *p*-value <0.01) according to a two-sided Fisher's exact test are highlighted in bold. Please see Methods and Source Data for details and exact *p*-values. Source data are provided as a Source Data file.

holoenzyme formation with regulatory proteins on the other hand.

**PP1c intrinsically recognizes basic motifs**. The analysis underlying Fig. 4a, b normalizes for the rate of natural occurrence of amino acids and allows visualizing amino acid preferences in an unbiased manner. We next sought to investigate whether the detected amino acid preferences would amount into a preferred sequence motif. Therefore, as complementary analysis to the above, we created a frequency matrix of PP1c/PP2Ac insensitive class I pS/pT sites and compared it to sensitive sites with a log2 fold change greater than −3 based on the phosphoproteomic data (Fig. 5a). For PP2Ac, no amino acid enrichment compared to

unaffected p-sites was obvious, and therefore the observed amino acid preferences did not directly translate into a motif, such as kinase target motifs. However, for PP1 we found that the observed preference for basic residues is most relevant in the context of Arg. In addition, we again noticed a statistically significant preference of PP1c but not of PP2Ac towards Arg in position −3. The sequence RXXpS is also found in the protein kinase A (PKA) and B (AKT) phosphorylation motifs[38], as well as in the 14-3-3-protein binding motif[39]. Therefore, we hypothesized that PP1c could have developed an intrinsic affinity to regulate phosphorylation events involving this biologically important motif, whereas PP2A might need association with regulatory subunits to gain specificity towards these motifs, as

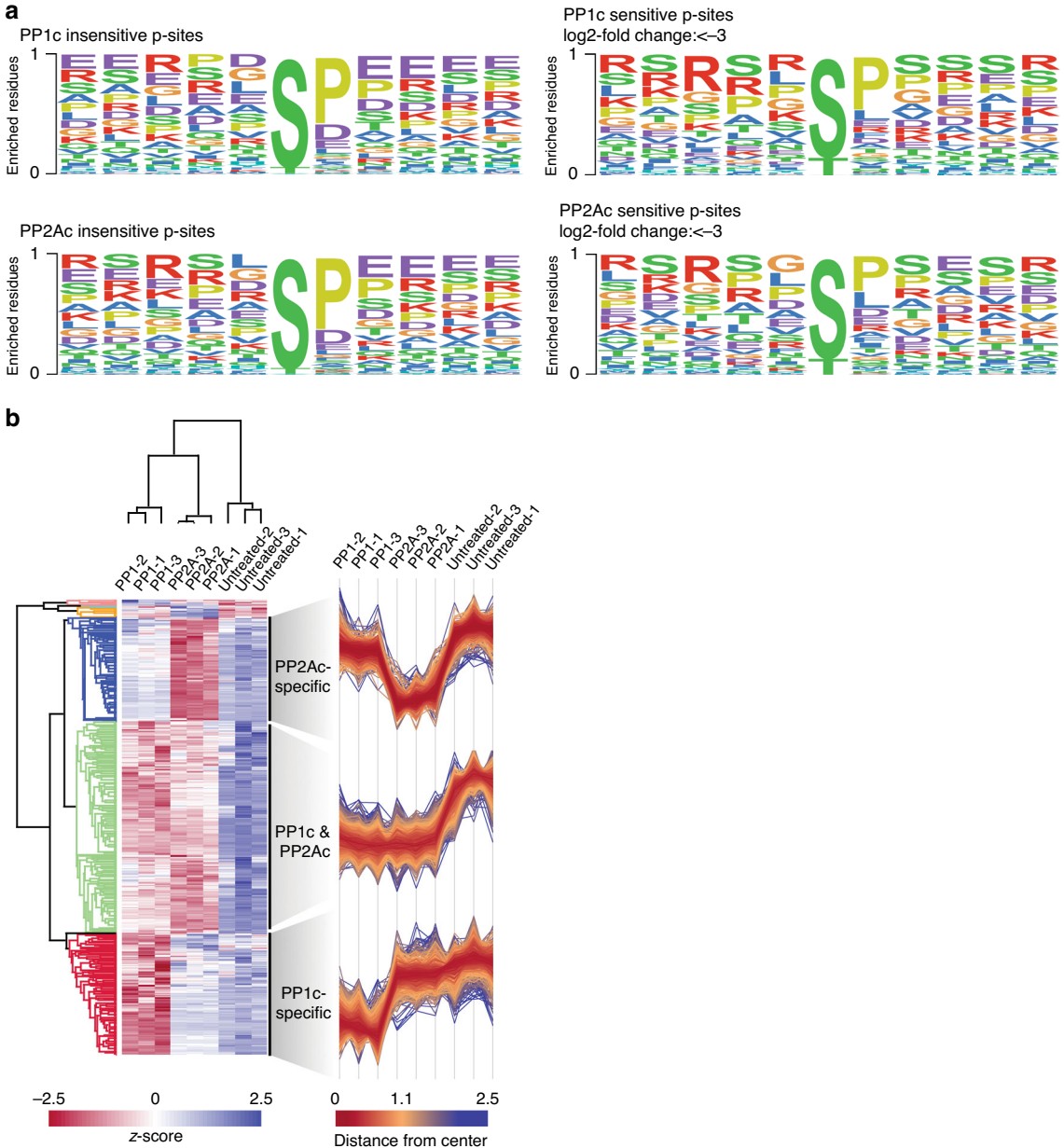

**Fig. 5 Analysis of PP1c/PP2Ac-affected p-sites.** The data are from the phosphoproteomic experiments (Supplementary Data 2; Supplementary Fig. 4b, c). **a** Motif analysis of PP1c/PP2Ac (in)sensitive class I sites. Color of single letter code represents biochemical properties with red = basic, purple = acidic. The enrichment of R in PP1c-sensitive p-sites over PP2Ac-sensitive sites was tested by a two-sided Fisher's exact test and was found to be statistically different (*p*-value 0.019). Source data are provided as a Source Data file. **b** Hierarchical clustering of p-sites after one-way ANOVA analysis using Perseus software. Color scale represents log2-fold change after z-transformation. The three largest clusters are depicted in detail on the right. Each line corresponds to one p-site. These data are also the basis for PP1c/PP2Ac-specific sites presented in Table 1.

PP2A holoenzymes are also known to regulate 14-3-3-binding sites[36,40–42]. To further strengthen the findings on this distinct intrinsic motif affinity for PP1c, we analyzed all p-sites by hierarchical clustering (Fig. 5b). This analysis revealed three major clusters, of which one contained 1361 sequences dephosphorylated by both PP1 and PP2A, the second 777 PP1c-specific sequences, and the third 663 PP2Ac-specific sequences (Supplementary Data 2). Accordingly, about 48% of the substrates were dephosphorylated by PP1c and PP2Ac, 28% were specific to PP1, and 24% specific to PP2A. Among p-sites differing with statistical significance between PP1c/PP2Ac, we found 12 RXXpS-containing p-sites on Ser/Arg-rich splicing factors (SRSFs) and

on six annotated 14-3-3-binding sites for PP1c, whereas only two sites in SRSFs and three 14-3-3-binding sites, but not containing a RXXpS motif, were identified as selective substrates for PP2Ac (Table 1). These findings corroborate the intrinsic affinity of PP1c for the RXXpS motif.

**PP1c dephosphorylates GAB2.** We next moved our attention to the RXXpS motif at residue pS210 of the docking protein GRB2-associated-binding protein 2 (GAB2), which was dephosphorylated by PP1c but not PP2Ac in the phosphoproteomic assay (Table 1, Supplementary Data 2). GAB2 was previously identified as 14-3-3-binding protein[43], but a link between this

**Table 1 Analysis of Arg/Lys-containing motifs among PP1c/PP2Ac-affected p-sites.**

| Gene name | Sequence window* | Residue | Local. Prob. | Note | Sensitivity |
|---|---|---|---|---|---|
| SRSF2 | KSRSR**S**KSPPK | 206 | 1.00 | S/R-rich splicing factor | PP1c |
| SRSF8 | KRPPK**S**PEEEG | 273 | 1.00 | S/R-rich splicing factor | PP1c |
| SRSF6 | RSQSR**S**NSPLP | 301 | 1.00 | S/R-rich splicing factor | PP1c |
| SRSF2 | RSRSR**S**PPPVS | 191 | 1.00 | S/R-rich splicing factor | PP1c |
| SRSF2 | RSRSR**S**RSPPP | 189 | 1.00 | S/R-rich splicing factor | PP1c |
| SRSF8 | SRYSR**S**PYSRS | 158 | 1.00 | S/R-rich splicing factor | PP1c |
| SRSF7 | SRYFQ**S**PSRSR | 192 | 1.00 | S/R-rich splicing factor | PP1c |
| SRSF5 | RSVSR**S**PVPEK | 233 | 1.00 | S/R-rich splicing factor | PP1c |
| SRSF8 | SPYSR**S**PYSRS | 163 | 1.00 | S/R-rich splicing factor | PP1c |
| SRSF1 | VDGPR**S**PSYGR | 199 | 1.00 | S/R-rich splicing factor | PP1c |
| SRSF10 | RSRSR**S**FDYNY | 133 | 1.00 | S/R-rich splicing factor | PP1c |
| SRSF5 | KSRSV**S**RSPVP | 231 | 0.99 | S/R-rich splicing factor | PP1c |
| SRSF4/5/6 | VENLS**S**RCSWQ | 113;117;119 | 0.97 | S/R-rich splicing factor | PP2Ac |
| SRSF11 | AAGLV**S**PSLKS | 207 | 1.00 | S/R-rich splicing factor | PP2Ac |
| GAB2 | NA**RS**A**S**FSQGT | 210 | 0.94 | 14-3-3 interaction site | PP1c |
| RAF1 | YQ**RRAS**DDGKL | 43 | 1.00 | 14-3-3 interaction site | PP1c |
| SOS1 | RR**R**PE**SAP**AES | 1161 | 1.00 | 14-3-3 interaction site | PP1c |
| CDK16 | IN**K**RL**SLP**ADI | 119 | 1.00 | 14-3-3 interaction site | PP1c |
| TBC1D4 | RG**R**LG**S**VDSFE | 588 | 1.00 | 14-3-3 interaction site | PP1c |
| USP8 | LK**R**SY**SSP**DIT | 718 | 0.99 | 14-3-3 interaction site | PP1c |
| HSF1 | PSPPQ**S**PRVEE | 307 | 1.00 | 14-3-3 interaction site | PP2Ac |
| HSF1 | KEEPP**S**P**P**QSP | 303 | 1.00 | 14-3-3 interaction site | PP2Ac |
| YAP1 | VRAHS**S**PASLQ | 127 | 0.97 | 14-3-3 interaction site | PP2Ac |

*Bold letters highlight the phosphorylation site and residues of the 14-3-3 consensus motif.
Serine/Arginine-rich splicing factors with the sequence window of −5 to +5 relative to pSer significantly decreased exclusively by PP1c or PP2Ac (top). These data are based on Fig. 5b and Supplementary Data 2. Known 14-3-3 interaction sites (annotated in PhosphositePlus) exclusively dephosphorylated by PP1c or PP2Ac.

interaction and dephosphorylation by PP1 has not yet been reported. Upon inhibition of endogenous PPPs in cells by Calyculin A, the p-level on this site was increased, demonstrating a role for PPPs in regulation of GAB2 on pS210. Incubation of the lysate with recombinant PP1c led to a decrease in the p-level (Fig. 6a). This suggests that pS210 of GAB2 is a substrate candidate of PP1c. We also demonstrated that the interaction of GAB2 with overexpressed GFP-14-3-3β was strongly increased upon inhibition of endogenous PPPs by Calyculin A, and decreased upon addition of recombinant PP1c in a PP1c–concentration-dependent manner, but not if recombinant PP1c was inhibited prior to treatment (Fig. 6b), suggesting altogether a specific interaction between GAB2 and PP1c.

Previously, GAB2 was shown to be sequestered in the cytosol by 14-3-3 and to be recruited to the plasma membrane upon epidermal growth factor receptor (EGFR) stimulation, which leads to the disruption of the interaction between 14-3-3 and GAB2[43]. The phosphatase responsible for dephosphorylating GAB2 and releasing it from 14-3-3 is not known[43]. Since PP1c activity led to the disruption of the GAB2-14-3-3 interaction in vitro (Fig. 6b), we therefore aimed to study the effect of PP1 activity on GAB2 localization in cells. For studying PPP–substrate interactions in cells, phosphatase-specific activation or inhibition within minutes is needed to reduce the risk of indirect and pleiotropic effects due to the activation of downstream and feedback cascades. Therefore, chemical tools instead of classical molecular biology techniques are required. However, explicitly specific inhibitors of PP1 are still not available[10,17]. To study the impact of PP1 activity on GAB2 localization, we therefore made use of a selective chemical PP1 modulator (PP1-disrupting peptide, PDP-*Nal*), which is a 23mer peptide that liberates active PP1c from holoenzymes in cells[36]. To test this relationship in different cell types, we not only considered HeLa Kyoto cells but also cancer cells derived from other epithelial tissues (Caco-2 BBe1, SW-480). Using live-cell microscopy of starved, mKate2-GAB2-transfected HeLa Kyoto, Caco-2 BBe1 and SW-480 cells

we demonstrated that EGF treatment led to GAB2 recruitment to the plasma membrane, as reported[43] (Fig. 6c, Supplementary Fig. 8a, b). Activating PP1c by PDP-*Nal* treatment partially recapitulated this effect within 5 min (HeLa Kyoto) or 12 min (Caco-2 BBe1, SW-480), whereas a control peptide only differing from PDP-*Nal* by substitutions of Val and Phe to Ala in the PP1-interacting RVxF-motif (PDPm-*Nal*[36]) showed no effect (Fig. 6c, Supplementary Fig. 8a, b, Supplementary Movies 1–9). This was confirmed by single-cell analysis of the imaging data using the cell segmentation strategy depicted in Fig. 6d and quantifying the signal at the cells' edge versus cyto/nucleoplasmic signal (Fig. 6e, Supplementary Fig. 8, see Eq. (1) in methods). These results show that liberating active PP1c with PDP-*Nal* leads to GAB2 recruitment to the membrane, which could be due to the loss of 14-3-3 binding as seen in vitro upon PP1c treatment (Fig. 6b). Together, our results provide initial evidence for PP1 as potential GAB2 phosphatase and give the intrinsic affinity of PP1c for GAB2 carrying the RXXpS motif at S210 a possible biological relevance.

**PP1c can regulate 14-3-3 complexes.** Since we observed an intrinsic affinity of PP1c towards the basic 14-3-3 motif, we were interested to see if this could be of broader significance and therefore sought to identify 14-3-3 targets regulated by PP1c using co-IP with MS read-out. To this end, we subjected GFP-14-3-3β–expressing cells to calyculin A treatment to inhibit endogenous PPPs and mixed these lysates with recombinant PP1c. Cells expressing GFP and untreated GFP-14-3-3β-expressing cells were used as controls. Proteins bound to immunoprecipitated GFP-14-3-3β were analyzed by LC-MS/MS using label-free quantification (LFQ) (Fig. 7a, Supplementary Fig. 9a). When comparing GFP-only control samples to untreated GFP-14-3-3β-expressing cells we identified 108 interacting proteins. Within these we noted a clear enrichment not only of known 14-3-3-interacting proteins, but of all seven mammalian 14-3-3 isoforms,

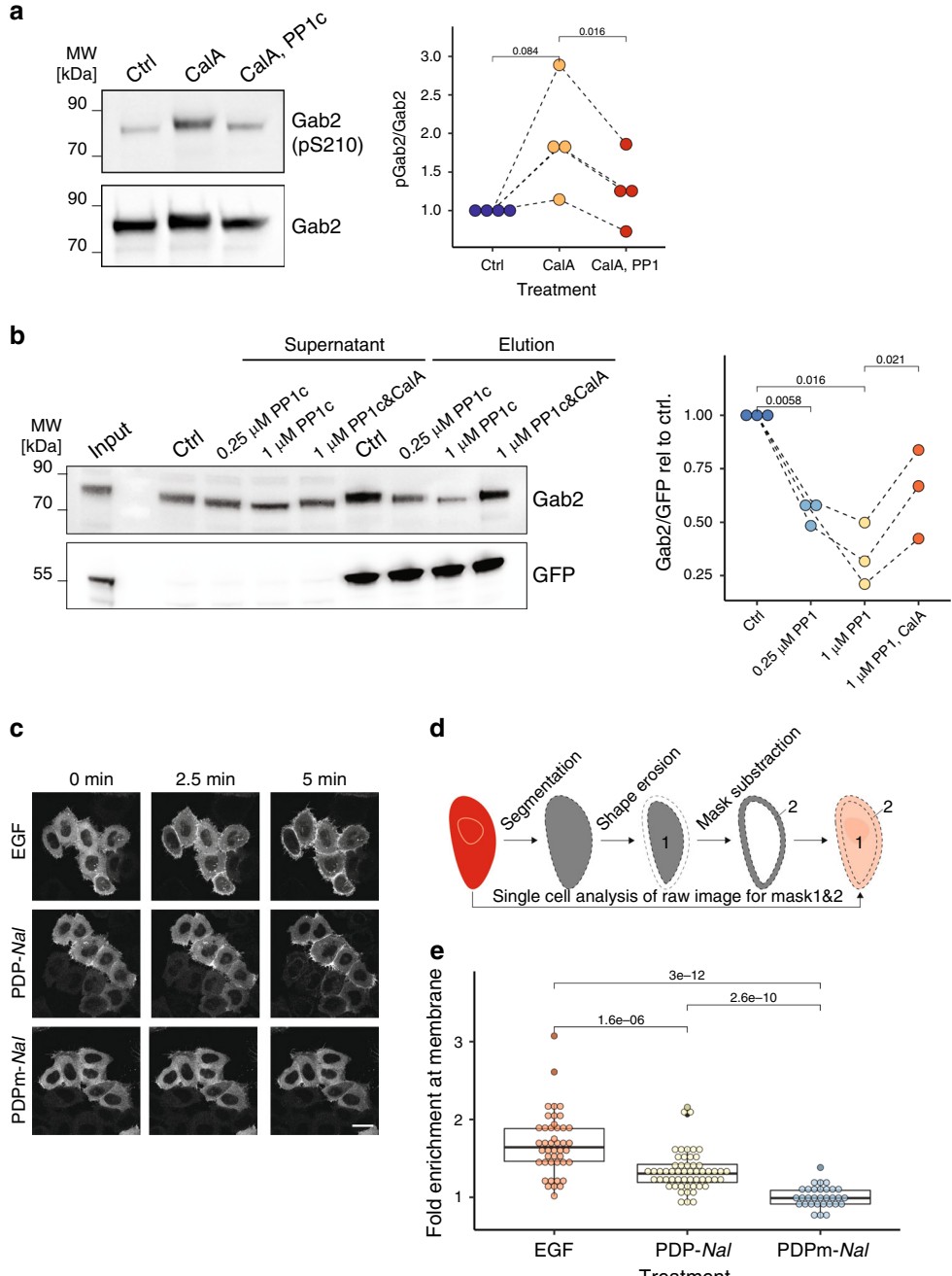

**Fig. 6 GAB2 is regulated by PP1. a** Dephosphorylation detected by Western Blot. GAB2-pS210 signal relative to GAB2 is quantified on the right. Shown is one representative experiment out of three independent repeats (CalA: Calyculin A). Four independent repeats have been quantified on the right. *p*-Values are derived from a t-test (two-sided, paired). **b** Overexpressed GFP-14-3-3β is less associated with endogenous GAB2 upon PP1c treatment. Inhibition of PP1c with calyculin A (CalA) abolishes the effect. Quantification depicts results of three independent experiments with *p*-values obtained from t-tests (two-tailed, paired). For a, b: Samples were derived from the same experiment and gels/blots were processed in parallel. **c** Confocal microscopy images of pmKate2-Gab2 signal in live HeLa Kyoto cells are shown after the indicated treatments. Stimulus was added after 45 s, scale bar represents 25 μm. (see also Supplementary Movies 1-3) **d** Scheme of image analysis for calculating GAB2 enrichment at the plasma membrane. **e** Fold-enrichment of GAB2 at the plasma membrane, as determined by comparing 0 min and 5 min frames. Each point represents a single cell (n = 44/56/31 for EGF/PDP-*Nal*/PDPm-*Nal*). Statistics represent adjusted *p*-values of a two-sided Wilcoxon test. Boxplots highlight median and 1st/3rd quartile. Whiskers refer to the lowest and highest data points that are within a 1.5-fold interquartile range. Source data of Fig. 6a, b, and e are provided as a Source Data file.

suggesting this method isolates functionally intact heterodimeric 14-3-3 complexes bound to their partner proteins (Supplementary Fig. 9b). Treatment with PP1 led to a significant dissociation of 56 selectively binding proteins from 14-3-3 (Fig. 7b). These were mapped back onto our phosphoproteomic dataset to identify at least one PP1c-regulated phosphorylation site in 35 of these proteins (Fig. 7c, Supplementary Data 3), of which 21 were dephosphorylated at the RXXpS motif, and additional 11 had at least one Arg or Lys at positions −1 to −5 adjacent to the p-site. Thus, PP1c disrupted 52% of the 14-3-3-protein interactions, and we saw in our phosphoproteomics study that 57% of the disrupted proteins carry one or more basic amino acids N-terminal

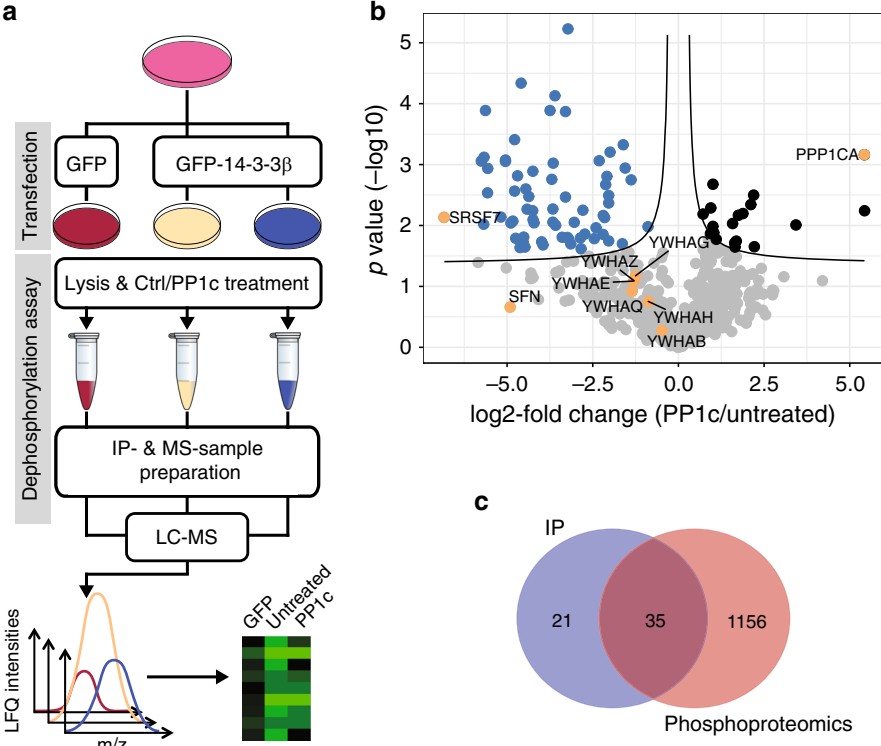

**Fig. 7 PP1 is a major regulator of 14-3-3 binding proteins. a** Workflow of GFP-14-3-3β IP and subsequent MS analysis to identify 14-3-3-interacting proteins regulated by PP1. **b** Volcano plot of pairwise comparison between untreated/PP1c-treated samples (two-tailed Student's *t* test, FDR 0.05). Proteins labeled in blue showed significant dissociation from 14-3-3 upon PP1c treatment and in orange represent SRSF proteins also identified in Table 1, 14-3-3 isoforms, and PP1 (added as recombinant protein). **c** Venn diagram showing overlap between the 56 proteins significantly dissociating from 14-3-3 upon PP1c treatment (but not binding GFP alone) compared to significantly dephosphorylated proteins (1967 p-sites on 1191 proteins) found in the PP1 phosphoproteomic experiments (t-test untreated vs. PP1 treated, see also Supplementary Fig. 4b and Supplementary Data 2, 3).

to the p-site that is dephosphorylated by PP1. Together, these data confirm the trends of intrinsic specificity toward basic motifs observed in the PLDMS and phosphoproteomic approaches, and strongly suggest that in PP1 the specificity for these biologically important motifs is ingrained in the catalytic subunit.

## Discussion

From a technological perspective, our PLDMS assay offers an approach for the use of large, highly complex peptide libraries to test effects of PTMs. Peptide libraries can be produced following either biological[44,45] or chemical principles[28–30,46]. However, biological approaches for peptide screening, with phage display being the most popular, usually depend on stable peptide binding with rather high affinities, and are complicated to set up to study PTMs because these are not genetically encoded[30,44,45]. If used for enzymatic reactions, antibodies can detect for example the phosphorylated substrate[30,44,45]. However, phosphatases remove a phosphate group, requiring a negative read-out of the dephosphorylated substrate[28,30,44]. Therefore, in biological libraries full phosphorylation would have to be ensured first before reliable results for dephosphorylation can be measured[44,45]. Thus, such biological approaches cannot be applied reliably and in a defined manner to study enzymes that remove a PTM like the PPPs. Chemical synthesis can follow parallelized, more cost-intensive synthetic approaches in which each well or spot on a microarray contains a distinct phosphopeptide for analysis[28–30]. Again, microarray technology requires a negative read-out similar to phage libraries[28,30,44]. On the other hand, randomized libraries can be synthesized effectively in a single tube setup, yielding a highly complex mixture of peptides. Such randomized libraries

can be obtained by a single synthesis but need advanced MS methodology for assay deconvolution after treatment[31].

We demonstrate that by tailored library design and optimization of an isokinetic mixture (that is a mixture of side-chain-protected amino acids with equal coupling efficiency) it is possible to cover >2000 unique peptide sequences in a single synthesis that also fulfill the criteria for robust LC-MS/MS analysis of PTMs. Furthermore, we show that rigorous control of library complexity such as length, number of randomized positions, and p-site in a defined position, allows a degree of empirical data filtering which would not be possible from proteomic datasets. Due to the fact that IsobarQuant/Mascot matches can be filtered that are impossible to be obtained from our synthetic routes, we were not only able to separate wrong from correct peptide populations, but decrease the FDR significantly below 5%. Moreover, since peptide synthesis guarantees pSer/pThr to be incorporated exclusively in position 0, our filtering strategy overcomes a major challenge of MS analysis, namely PTM-site prediction. Statistical significance obtained from PLDMS data clearly outcompeted statistics on our phosphoproteomic datasets due to the advantages discussed above. Combining the two methods then allowed to narrow down the significant changes and to identify potential biological explanations from using peptides such as the strong disfavor of Pro at position +1 for PP2A. Given that our strategy overcomes many limitations and offers many advantages, we expect that it will have a broad applicability to different classes of enzymes, particularly of enzymes that remove a PTM.

On the biological side, the lack of knowledge of phosphatase substrate networks as compared to kinase substrate networks arises from the hurdles imposed by protein evolution: pSer/pThr-

kinases diversified at the level of the catalytic protein, whereas pSer/pThr-specific PPPs conserved catalytic subunits and diversified on the level of regulatory subunits[7,9]. Since PP1 and PP2A exist in cells as holoenzymes, substrate recruitment happens at this level. Nevertheless, some evidence suggested already that amino acids in close proximity to the p-site could influence phosphatase activity[20–22], implying a multi-layer specificity system for p-site dephosphorylation of PPP holoenzymes.

Without questioning the importance of the holoenzyme in substrate recruitment and specificity, we show that the intrinsic priming of PP1c and PP2Ac for p-site selection after substrate recruitment is distinct. Whether all preferences that we found are relevant in a biological context is not a trivial question and will require future studies. We focused here on the fact that in contrast to PP2Ac, PP1c preferentially dephosphorylates at sites where the N-terminal amino acids are positively charged. Biological relevance on the holoenzyme level for this result becomes apparent when analyzing available crystal structures. As introduced, the catalytic cleft of both phosphatases is surrounded by three grooves[11,12,47]. A peptide stretch within a substrate protein was suggested to bind the phosphatase by a combination of two out of these three grooves[13]. Interestingly, the acidic groove close to the active site is much less pronounced in PP2Ac[47,48](Fig. 8a), since the acidic residues Asp220, 253, 277 and Glu218, 252, 256, are exclusively found in PP1 (Fig. 8b). Furthermore, crystallization experiments of PP1c with regulatory subunits have demonstrated that MYPT1[49] further extends the acidic pocket, and spinophilin/neurabin[50] and PNUTS[51] even occlude the C-terminal groove, thereby forcing substrates towards the hydrophobic/acidic pockets. This supports that in a cellular system a subset of PP1 substrates makes active use of the acidic groove for p-site selection after recruitment.

How can PP2Ac still dephosphorylate basic motifs? Our inspection of PP2A complexes suggests that this substrate specification is achieved in a differential manner by regulatory subunits. Earlier structural studies of the PP2A holoenzyme with the subunit B55 revealed that a highly acidic loop between Glu81 and Glu94, with six out of 14 amino acids being Asp or Glu, binds closely to the catalytic site. Distances between the catalytic site metal $Mn^{2+}$(502, labeled with * in Fig. 8b) and $C\gamma/\delta$ of different Asp or Glu, in this structure (PDB ID 3DW8) are 34.6 Å (Glu81), 27 Å (Glu83), 17.7 Å (Asp85), 28.2 Å (Glu91), 30.6 Å (Glu93), 34.2 Å (Glu94) (Supplementary Fig. 10)[52]. In the case of the B56 subunit, a loop containing Asp109, Glu111, 112, 113, and 114 is docking even closer to the catalytic site, locating the aforementioned acidic side chains 18/20.5/13.9/21.4/20 Å from the $Mn^{2+}$ in the catalytic cleft (Fig. 8c, distances between $Mn^{2+}$(502) and $C\gamma/\delta$ of Asp or Glu, respectively, in PDB ID 2NPP)[48]. Of note, during the revision process of this manuscript an independent study confirmed the role of B56 in targeting basic p-sites[53]. Thus, PP2A requires the regulatory proteins for the recognition of basic motifs, whereas PP1 can recognize them through the catalytic subunit (Fig. 8d). Such data are not only interesting for the basic understanding of PP1 and PP2A catalytic activity and substrate recognition, but also for the design of selective drugs to target them. The interfaces that would be targeted to disrupt the interactions of PP1 and PP2A with their substrates are different (direct contact to catalytic subunit versus to the holoenzyme), leading to targeting possibilities for inhibitor design.

In agreement with previous observations[22,24–27,54,55], we show that PP1c and PP2Ac dephosphorylate pThr and pSer, but indeed globally prefer the less abundant pThr. Of note, by analyzing >10,000 different sequences we demonstrate in a comprehensive, unbiased manner that the contribution for this preference is introduced by the catalytic subunits, not by regulatory subunits.

Importantly, the latest study[53] showed that in case of PP2A-B56 this intrinsic preference is overruled through the subunit, whereas they confirmed the preference for pThr for PP2A-B55, emphasizing the important interplay between the specificities of the catalytic and regulatory subunits. We show that the dephosphorylation efficiency ($k_{cat}/K_m$) of the catalytic subunits with pThr contributes to the preference (Fig. 8e). The biological relevance of this general finding in the context of holoenzymes is demonstrated by the observation that in the specific process of mitosis, PP2A holoenzymes dephosphorylate pSer more slowly than pThr and this enables the dephosphorylation kinetics necessary to exit mitotis[24–27]. Furthermore, recent findings show a faster dephosphorylation of pThr over pSer for PP2B/Calcineurin based on the analysis of the PP2B: $Na^+/H^+$ exchanger 1 (NHE1) interaction[56]. While this is only a single experimentally validated case on the protein level, it extends the pThr preference to another PPP member. Moreover, this study also discovered an active site recognition sequence pS/TXXP for the PP2B:NHE1 interaction, and implied that this could be a general active site recognition motif by screening 48 interaction partners[56]. Applying our method to PP2B would strongly substantiate the data of the recent study, showing the broad applicability of the strategy.

Another major gain of our strategy is that it resulted in the identification of a large set of PP1/PP2A substrate candidates. Accordingly, we revealed a connection of PP1c substrate candidates to the 14-3-3 interactome and identified the 14-3-3 binding site at pS210 of GAB2 as a site regulated by PP1. GAB2 is a scaffolding protein directly linking receptor tyrosine kinase stimulation to the cancer-relevant ERK and PI3K-AKT pathways[43,57]. PP1 activation using PDPs was sufficient for GAB2 recruitment to the plasma membrane, even in the absence of stimulus. This effect was observed in three distinct cancer cell lines derived from epithelial tissues in cervix and colon. We therefore suggest that the dephosphorylation of pS210 of GAB2 may play an important role in counteracting GAB2 sequestration in the cytosol. Therefore, inhibition of the GAB2/PP1 interaction might represent an approach for inhibition of erroneous cell proliferation downstream of EGFR in epithelial cancer.

In summary, we present an MS-based strategy for characterizing PPP-substrate specificities, which enabled us to reveal intrinsic specificities of PP1c and PP2Ac, and identify a large set of substrate candidate p-sites, including those with a direct link to 14-3-3-protein function. The application of the strategy to other phosphatases and potentially further enzymes promises a large gain in the understanding of enzyme-substrate recognition around the PTM site, as well as the identification of multiple substrate candidates.

## Methods

**Cell culture**. HeLa Kyoto cells were obtained from the BIOSS Signalling Factory cell line repository (University of Freiburg), Caco-2 BBe1 cells from ATCC (ATCC CRL-2102), and SW-480 cells from the European Collection of Authenticated Cell Cultures (ECACC 87092801). Cells were cultured at 37 °C in a humidified incubator under 5% $CO_2$. The cells were grown in GlutaMax Dulbecco's modified Eagle's medium (DMEM, Gibco) supplied with 1 g/L glucose, 10% (v/v) fetal bovine serum (FBS) and 1% (v/v) penicillin/streptomycin. Cells were passaged routinely when reaching 90% confluency by trypsinization and 10-fold dilution. All cells have been verified by PCR analysis of short tandem repeats by an independent external service provider (Labor f. DNA-Analytik, Freiburg, Germany) and were free of mycoplasma (GATC Biotech/Eurofins, Ebersberg, Germany). Please see the Source Data for certificates of analysis.

**Plasmids and cloning**. For all clonings, Phusion Polymerase (Thermo Fisher Scientific) was used following the recommended protocol for PCR setups recommended by the manufacturer, all restriction enzymes were FastDigest enzymes (Thermo Fisher Scientific). pcDNA3-FLAG-HA-14-3-3β was originally a gift from William Sellers (Addgene plasmid # 8999), but sequenced and corrected for a C-terminal extension diverging from the canonical amino acid sequence with primers 5′-GGGAGACCCA AGCTTACCATGGAC-3′ (fwd) and 5′-CCGCCGAAGCGGCCGCTATTAGTTCT

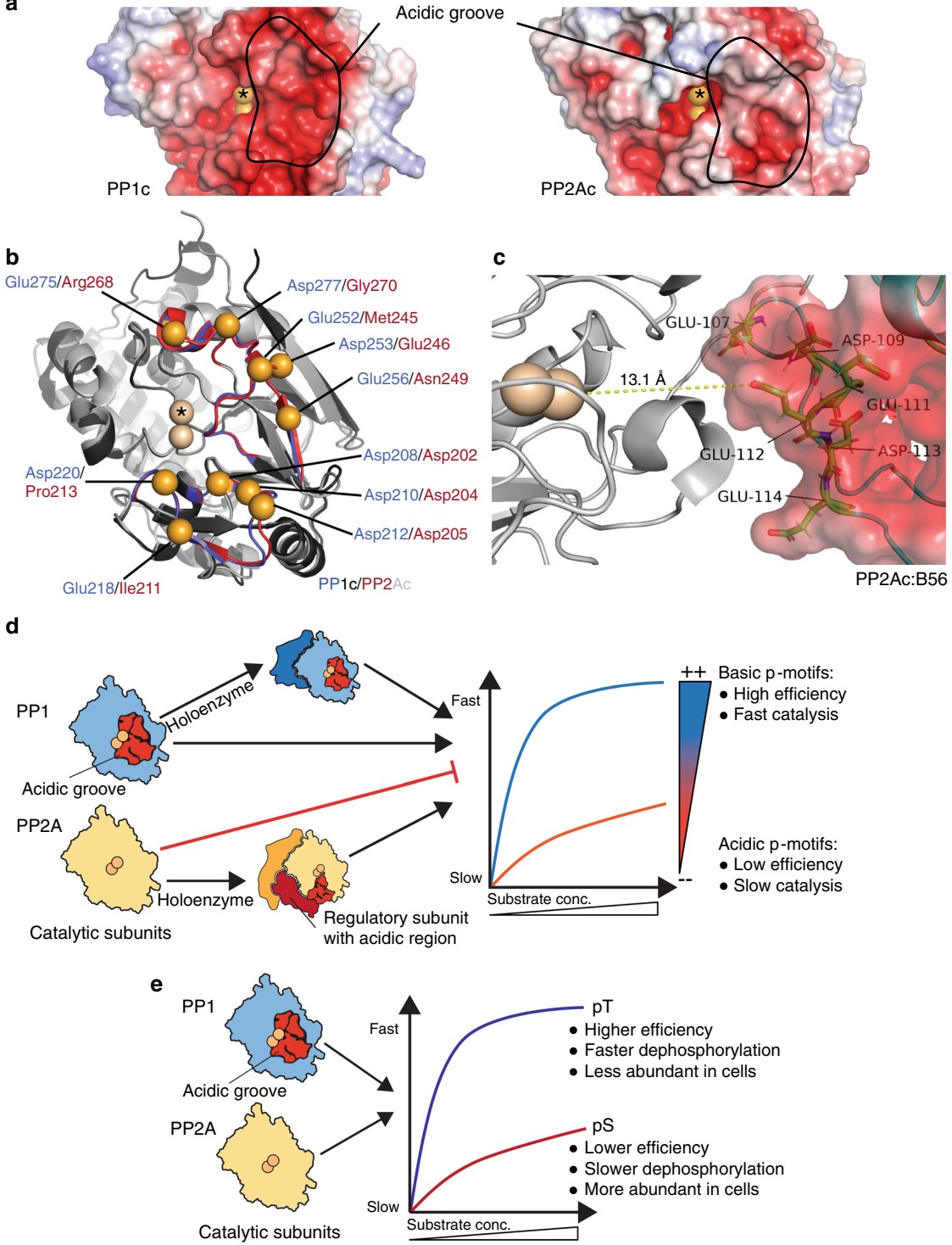

**Fig. 8 Relating the amino acid preference of PP1c and PP2Ac to holoenzymes. a** The acidic groove is much more pronounced in PP1c compared to PP2Ac. Structures for PP1 catalytic subunit alpha isoform (PDB ID 3EGG, chain A)[50] and PP2A catalytic subunit alpha isoform (PDB ID 4I5L, chain C)[69] were retrieved from www.pdb.org (accessed 25 Oct 2019) and inspected in PyMOL v2.3.3. * marks the catalytic cleft containing two $Mn^{2+}$ ions. Color coding between −5 (red) to +5 (blue) $K_bT/e$. **b** Aligned structures of PP1c and PP2Ac (see above for details) highlighting residues which determine differences in acidic properties (orange spheres). The sequence alignment to determine corresponding residues was carried out in Needle (EMBOSS). Color coding of structures: PP1c in blue/black, PP2Ac in red/gray. Residues constituting the acidic groove have previously been defined by Zhang et al.[70] **c** Interaction of PP2Ac with B56/B′/PR61 based on the crystal structure of the trimeric holoenzyme with the scaffolding subunit A (PR65) derived from PDB ID 2NPP[48]. Negatively charged amino acids within the binding regions of B56 to PP2Ac are highlighted. **d** The holoenzyme of PP1 has an intrinsic preference for basic motifs due to the composition of the acidic groove of the catalytic core protein. PP2A holoenzymes are *per se* not primed towards amino acid sequence features of 14-3-3 or PKA motifs, but need to associate to regulatory subunits such as B56 carrying acidic patches to achieve basophilic sequence specificity. **e** Both, PP1 and PP2A holoenzymes have a preference for pT due to a higher catalytic efficiency of their respective catalytic subunits towards pT over pS.

CTCCCTCCCCAGCGTC-3′ (rev) and reinserted with HindIII/NotI before use. For pEGFP(C1)-14-3-3β, YWHAB was subcloned into pEGFP(C1) by attaching BglII/ApaI sites with primers 5′-GTTGTTGGAAGATCTACAATGGATAAAAGTGAGCTGGTACAGAAAG-3′ (fwd) and 5′-ATAGAATAGGGCCCTCTAGATGCATGC-3′ (rev). For pmKate2-N-GAB2 (i.e. C-terminal tagging of GAB2), GAB2 was amplified from pMIG-GAB2 (gift from T. Brummer, University of Freiburg) with primers 5′-CGAGGTTCTAGACTCGAGATG-3′ (fwd) and 5′-GGTAGGGATCCCGCAGCTTGG-3′ (rev) to attach XhoI/BamHI sites by PCR and inserted using the afore-mentioned sites. For pCMV-GAB2-3Flag, 3Flag was amplified from pCMV-3Tag-A1 and and stop codons and BamHI/HindIII sites were attached by PCR with primers 5′-GCCAAGCTGGGATCCGATTACAAGGATGACGACGATAA-3′ (fwd) and 5′-GGTCCGAACAAGCTTTCATTTATCGTCATCATCTTTGTAGTCC-3′ (rev). GAB2 was amplified from pMIG-GAB2 and NotI/BamHI sites were attached using the primers 5′-GGATCCGGAGCGGCCGCGAGATGAGCGGCGGC-3′ (fwd) and 5′-ATCCTTGTAATCGGATCCCAGCTTGGCACC-3′ (rev). Fragments were fused by PCR and the flanking primers and inserted into pCMV digested NotI/HindIII sites. pET-PPP1R2 was a gift from the Bollen Lab (KU Leuven). The PPP1R2 sequence (Inhibitor-2, I-2, from rabbit) was amplified using primers 5′-GTTGTTGGACCATGGCGGCCTCGACGGCC-3′ (fwd) and 5′-ATAAGAATGCGGCCGCTTATCACTATGAACTCTGTGATTTGTTTTGTCG-3′ (rev) to attach NotI/NcoI sites and subcloned into pETM-20. All plasmids were checked for sequence integrity by Sanger sequencing before further use.

**Transfection**. For transient transfections, cells were plated one day before in order to reach 40–50% confluency on the day of transfection. All transfections were carried out using FuGene HD transfection reagent (Promega) and followed the manufacturer's instructions using OptiMEM (Gibco) and a ratio of plasmid:transfection reagent (w/v) of 1:6. For immunoprecipitation (IP) experiments of pCMV-GAB2-3Flag followed by Western blotting or MS analysis, 5 μg of DNA were used for transfection in a cell culture plate with a diameter of 10 cm and expressed for 24 h. For transfections of pcDNA3-FLAG-HA-14-3-3β and pEGFP (C1)-14-3-3β, 2.5 μg were transfected, again for 24 h. For live-cell microscopy of pmKate2-GAB2, 100 ng/well were transfected in cells plated in 8-well μ-slides (ibiTreat-coating, Ibidi).

**Library design for PLDMS experiments**. In order to limit the complexity of the library and increase data quality during analysis, we anticipated 5000–6000 theoretical peptides per library as an optimal complexity, since this would result in 10,000-12,000 theoretical masses for peptides upon dephosphorylation, which would still lead to redundant measurements of the same peptide during a several-hour LC-MS/MS run. To decide how to reduce the complexity, known PP1 and PP2A protein substrates[8] were analyzed. This revealed that the most influence on PP1 substrate recognition is within positions −4 to +3 relative to pSer/pThr. For PP2A, no preference was found. Based on this analysis, each library peptide was designed to consist of 10 amino acids with a pSer/pThr in position 0 (5th amino acid in the peptide sequence, see Fig. 1b) and a C-terminal Ala-Lys. Using 14 different potential amino acids instead of the naturally occurring 20 at 3 permutated positions, as well as either pSer or pThr in position 0, would result in $14^3 \times 2 = 5488$ different theoretical 10mer peptide sequences per synthesis, matching our envisaged complexity (for comparison, using all 20 amino acids would increase the complexity of each library to $20^3 \times 2 = 16,000$). In addition to allowing high data quality, using 14 instead of 20 amino acids would also increase the quality of the synthesis of the peptides: a lower complexity of the amino acid mixtures required for synthesis leads generally to better yields, more even distribution of amino acids in the positions, and less by-products, in turn increasing the MS measurement quality.

To cover these residues within the limits for complexity discussed above, positions −4 to −1 (a total of 4 positions) required 4 separate libraries with 3 randomized positions and 1 fixed position (Ala) each, and a single library was necessary to cover the 3 randomizations in position +1 to +3 (a total of 3 positions) (Fig. 1b). Furthermore, we decided for Leu over Ile to reduce database issues due to identical masses. Thus, we used four N-terminal libraries (Nterm libraries), named x1_1, x1_2, x1_3, and x1_4, which are composed of three permutated positions randomly containing A/E/F/G/H/K/L/P/Q/R/S/T/V/Y and a fixed Ala in the position −4, −3, −2, or −1, respectively. C-terminal positions of the Nterm libraries consisted of the sequence AAAAK. We chose Ala for the fixed positions as a neutral amino acid and spacer from Lys that should have no impact (as judged from known PP1 substrates), and the C-terminal Lys for the benefits of an additional charge for MS analysis. To analyze the role of different amino acids C-terminal of the phosphorylated site, we synthesized a library following the same rationale with three permutated positions C-terminally (Cterm library) next to the phosphorylated site using the amino acids A/D/E/F/G/K/L/N/P/Q/R/S/T/V, followed by the C-terminal Ala-Lys, and with four Ala N-terminal of pSer/pThr. As a result from the design based on known protein substrates (see above), 12 amino acids were in common for both N- and Cterm libraries, and 2 each were unique for either N- or Cterm libraries. This resulted in a setup of 19,710 (4 × 5488 minus redundant sequences) different theoretical phosphopeptides in the Nterm libraries and 5488 different phosphopeptides in the Cterm library, and a total substrate variety of 25,196 different theoretical phosphopeptides.

**Materials for peptide synthesis**. All amino acids and resins were purchased from Novabiochem (Merck, Darmstadt, Germany). All other synthetic reagents were obtained from Novabiochem, Sigma-Aldrich (St. Louis, Missouri, USA), or Carl Roth (Karlsruhe, Germany). Peptide synthesis was performed on a MultiPep RSi peptide synthesizer (Intavis Bioanalytical Instruments AG, Köln, Germany). Peptides were purified using a 1260 Infinity II Prep HPLC system (Agilent Technologies, Santa Clara, California, USA) with a VP 250/10 NUCLEODUR 100-5 C18 ec column (Macherey-Nagel, Düren Germany) running a general gradient of 10% to 50% Acetonitrile (ACN) in $H_2O$ and validated using a 1260 Infinity I HPLC System coupled to a 6120 Quadrupole LC/MS (Agilent Technologies, Santa Clara, California, USA) with an EC 250/4 NUCLEODUR 100-5 C18 ec column (Macherey-Nagel) also running a general gradient of 10% to 50% ACN in $H_2O$ and a Microflex LT MALDI (Bruker, Bremen, Germany).

**Peptide and library synthesis**. Peptides were synthesized following fluor-enylmethyloxycarbonyl- (Fmoc) solid-phase peptide synthesis (SPPS). Wang resin or pre-coupled H-Lys(Boc)-2-ClTrt resin was used for all peptides and libraries. Resins were swollen for 20 min in N,N-dimethylformamide (DMF) prior to synthesis and then transferred to the peptide synthesizer. One round of synthesis was performed for each amino acid in the peptide sequence. A synthetic round consisted of double coupling to an Fmoc and side-chain-protected amino acid residue, followed by a capping step with acetic anhydride ($Ac_2O$), and then removal of the Fmoc group with piperidine in preparation for the next round of coupling. Coupling reactions were performed by adding Fmoc-protected amino acids (4 eq.), O-benzotriazole-N,N,N′,N′-tetramethyl-uronium hexafluorophosphate (HBTU) (4 eq.), N-hydroxybenzotriazole hydrate (HOBt) (4 eq), and N-methylmorpholine (NMM) to the resin in 1 mL DMF and reacting for 30–45 min. Capping was done by reaction with a 1 mL solution of 5% acetic anhydride $Ac_2O$ and 5% 2,6-Lutidine in DMF for 5 min. Fmoc deprotection was achieved by addition of a 20% piperidine in DMF solution to the resin, first for 3 min, then again for 8 min. Resins were washed with DMF between each step. After Fmoc removal of the final amino acid residue to reveal the free N-terminus, peptides and libraries were removed from the resin and fully deprotected in one step by shaking for 3–4 h in cleavage cocktail (92.5% trifluoroacetic acid (TFA), 5% triisopropylsilane (TIPS), and 2.5% $H_2O$). Peptide libraries were isolated by co-evaporation of the cleavage cocktail with toluene in vacuo and directly used for dephosphorylation assays and subsequent LC-MS/MS analysis. Individual peptides were precipitated in cold ether (−20 °C) and collected by centrifugation. Peptides were then validated and purified by MALDI, HPLC-MS, and Prep HPLC. For both sets of libraries, the following isokinetic ratio was used for the amino acid position including the phosphate group: phosphoserine and phosphothreonine (amino acid, mol %): Fmoc-Ser(PO (OBzl)OH)-OH, 50; Fmoc-Thr(PO(OBzl)OH)-OH, 50.

**Isokinetic amino acid mixture for library synthesis**. For the syntheses of the peptide libraries, variable positions were added to the peptide sequence. At these permutated positions, the resin was reacted with an isokinetic mixture of amino acids, in order to ensure an equal distribution of amino acids at that position. The following optimized isokinetic ratios were used to create the amino acid mixtures (amino acid, mol %). x1 Libraries: Fmoc-Ala-OH, 4.6; Fmoc-Arg(Pbf)-OH, 6.8; Fmoc-Glu(OtBu)-OH, 7.4; Fmoc-Gln(Trt)-OH, 11.0; Fmoc-Gly-OH, 9.9; Fmoc-His(Trt)-OH, 4.8; Fmoc-Leu-OH, 3.4; Fmoc-Lys(Boc)-OH, 8.6; Fmoc-Phe-OH, 3.4; Fmoc-Pro-OH, 5.9; Fmoc-Ser(tBu)-OH, 7.6; Fmoc-Thr(tBu)-OH, 13.2; Fmoc-Tyr(tBu)-OH, 5.6; Fmoc-Val-OH, 7.8. x2 Library: Fmoc-Ala-OH, 4.3; Fmoc-Arg (Pbf)-OH, 6.3; Fmoc-Asn(Trt)-OH, 10.2; Fmoc-Asp(OtBu)-OH, 6.8; Fmoc-Glu (OtBu)-OH, 6.8; Fmoc-Gln(Trt)-OH, 10.2; Fmoc-Gly-OH, 9.2; Fmoc-Leu-OH, 3.1; Fmoc-Lys(Boc)-OH, 8.0; Fmoc-Phe-OH, 3.1; Fmoc-Pro-OH, 5.5; Fmoc-Ser(tBu)-OH, 7.0; Fmoc-Thr(tBu)-OH, 12.3; Fmoc-Val-OH, 7.2. These ratios were based on a previously reported mixture[35] and optimized for our syntheses.

**Recombinant protein expression**. Recombinant PPP1CA was purified by the EMBL Protein Expression and Purification Core Facility (PEPCore) following an established protocol[58]: pTXB1-His-TEV-PP1α-intein was expressed in E.coli BL21Star(DE3)pRARE in LB broth including 1 mM $MnCl_2$ after induction with 50 μM isopropyl β-d-thiogalactoside (IPTG) at 16 °C overnight. The cell paste was lysed using an Emulsiflex homogenizer in 25 mM TRIS-Cl, pH 7.5 at room temperature (RT), 300 mM NaCl, 10% v/v glycerol, 30 mM imidazole, 0.2% v/v tween-20, 0.1 mM phenylmethylsulfonyl fluoride, EDTA-free protease inhibitor cocktail (Roche) and benzonase (Merck). Next, NaCl was added to the soluble fraction to a final concentration of 700 mM. Protein was then purified on a HisTRAP HP nickel column in 25 mM TRIS-HCl, pH 7.5 RT, 700 mM NaCl, 5% v/v glycerol, 30 mM imidazole, 0.2% v/v tween-20, followed by dilution and incubation with 50 mM β-mercaptoethanol (β-ME) and 1 mg TEV protease at 4 °C overnight for tag cleavage. Cleaved PP1c was then purified using chitin resin and on a Heparin HP column, equilibrated in 20 mM TRIS-Cl, pH 7.5 RT, 100 mM NaCl, 5 mM β-ME. Elution was carried out using a high salt gradient and protein was dialyzed into storage buffer. His-tagged recombinant PPP2CA ΔL309 was purchased from Cayman Chemical (#10011237). Recombinant Inhibitor-2 was obtained from pETM20-His-TEV-PPP1R2 after transformation, expression in BL21cells and purification[36]: Protein expression was carried out at 37 °C for 4 h in LB broth (containing

Ampicillin) after induction with 100 μM IPTG (final concentration). Next, the culture was harvested, resuspended in lysis buffer (50 mM Tris-HCl, 500 mM NaCl, 30 mM imidazole, 1x cOmplete Mini protease inhibitor cocktail from Sigma), and lysed by sonication on ice (3 cycles of 6 min). The lysate was cleared by centrifugation for 1 h at 6000 × g and 4 °C. The soluble fraction was then purified at 4 °C on an ÄKTA explorer system equipped with a HisTRAP HP nickel column (GE) equilibrated in lysis buffer. Protein was eluted in lysis buffer containing 250 mM imidazole and dialysed into 50 mM Tris-HCl, 500 mM NaCl for subsequent TEV cleavage (overnight). The next day, the aforementioned column purification was repeated using lysis buffer w/o imidazole and the flow-through containing the cleaved protein was collected, concentrated and supplemented with glycerol to a final concentration of 20% for long-term storage at −80 °C.

**Detection of phosphate release from peptides and libraries**. Prior to library dephosphorylation with subsequent MS analysis, as well as for the confirmation of MS results by directed peptide synthesis and dephosphorylation kinetics, the release of phosphate over time from peptides or libraries had to be monitored: The libraries and peptides were dissolved in 25% DMSO and 75% $H_2O$ at 10 mM. The EnzChek Phosphate Assay Kit (Thermo Fisher Scientific) was used to assess the kinetics/dynamics of the library or peptide dephosphorylation by PP1c and PP2Ac. The assay was carried out in a volume of 100 μL reaction buffer (20 mM Tris, 100 mM NaCl, 1 mM DTT, 0.15 units PNP, 0.2 mM MESG,) containing 100 μM phosphopeptide library or indicated concentrations of peptide and 25 nM phosphatase. Because of our goal to identify differences in selectivity, we used equimolar concentrations of the phosphatases. Using the same amount of enzyme based on the activity, i.e. in enzyme units, would falsify the outcome because the specificity we were interested in is based on interaction, i.e. binding, not turnover, and different concentrations of enzyme would have an effect on binding events. The change of absorbance was detected at 360 nm at 25 °C on a Synergy H4 microplate reader using the software Gen5. For the dephosphorylation assay with the library we aimed for a dephosphorylation rate between 30 and 50% of total dephosphorylation in order to have enough dephosphorylated peptides for significant analysis and to see substrate preference of the respective phosphatase without reaching saturation. For analysis of enzyme kinetics, raw data were further analyzed in GraphPad Prism v6.

To confirm that the time point for 30–50% library dephosphorylation could be reproduced on the same day and in the very same samples that would later be analyzed by LC-MS/MS, we applied Biomol green as a complementary phosphate detection assay to the EnzChek assay. 50 μL from the dephosphorylation reaction were removed before stopping the reaction. To this end, 100 μL of the Biomol green reagent (Enzo Life Sciences) were added at the same time point when HCl was added to the main reaction (see the following chapter). After incubation for 25 min the colorimetric change was measured at 620 nm at 25 °C on a Synergy H4 microplate reader. Values were compared to a standard curve prepared by using the phosphate-solution included in the kit and following the manufacturer's instructions. They were also compared to values for complete library dephosphorylation estimated either from the incubation of the library with FastAP alkaline phosphatase (Thermo Fisher Scientific) or by incubating library with PP1c for 1–2 h.

**Dephosphorylation assay for subsequent MS analysis**. A volume of 300 μL reaction buffer containing 100 mM NaCl, 20 mM Tris Cl, 1 mM DTT (pH 7.5), 100 μM phosphopeptide library, and 25 nM phosphatase was used for each sample. The sample was incubated for 2 min with PP1c and 4 min with PP2Ac at 25 °C, which resulted in 30–50% library dephosphorylation as determined by monitoring the release of inorganic phosphate as described above. The reaction was terminated by the addition of 13.5 μL 0.37% HCl, which caused a pH-shift from pH 7.5 to pH 3, followed by centrifugation for 5 min at 10,000 × g at 4 °C before subjecting samples to SEC.

**Library purification using SEC**. For LC-MS analysis, the treated/ctrl peptide libraries were purified by SEC. To this end, the sample was loaded onto a pre-equilibrated Superdex Peptide 10/300 GL column (GE Healthcare) with a flow rate of 0.5 mL/min, installed in an ÄKTA explorer system coupled to an automated fractionation system. A 0.1 M ammonium acetate buffer (pH 5) was used throughout the column equilibration and purification and the elution profile was recorded at 280 nm. The fractions of the library peak were collected, pooled together and concentrated by vacuum centrifugation overnight. The dried peptide libraries were then further analyzed by LC-MS/MS measurements.

**LC-MS/MS measurement for phosphopeptide library experiments**. The purified peptide libraries were reconstituted in 0.1% formic acid (FA) prior to MS measurement. Peptides were separated using an UltiMate 3000 RSLC nano LC system (Thermo Fisher Scientific) equipped with a trapping cartridge (Precolumn C18 PepMap 100, 5 μm, 300 μm i.d. size-exclusion chromatography 5 mm, 100 Å) and an analytical column (Waters nanoEase HSS C18 T3, 75 μm × 25 cm, 1.8 μm, 100 Å). Solvent A was 0.1% FA in LC-MS grade water and solvent B was 0.1% FA in LC-MS grade acetonitrile. After loading the peptides onto the trapping cartridge (30 μL/min of solvent A for 3 min), elution was performed with a constant flow of

0.3 μL/min using 120 min analysis time (with a 4–20%B elution, followed by an increase to 40%B, to 85% for washing and re-equilibration to initial conditions). The LC system was directly coupled to a Q Exactive Plus mass spectrometer (Thermo Fisher Scientific) using a Nanospray-Flex ion source and a Pico-Tip Emitter 360 μm OD × 20 μm ID; 10 μm tip (New Objective). The mass spectrometer was operated in positive ion mode with a spray voltage of 2.3 kV and capillary temperature of 275 °C. Full scan MS spectra with a mass range of 275–1000 m/z were acquired in profile mode using a resolution of 70,000 (maximum fill time of 30 ms or a maximum of 3e6 ions (automatic gain control, AGC)). Fragmentation was triggered for the top 10 peaks with charge 1 to 3 on the MS scan (data-dependent acquisition) with a 1 s dynamic exclusion window (normalized collision energy was 32), and MS/MS spectra were acquired in profile mode with a resolution of 35,000 (maximum fill time of 120 ms or an AGC target of 2e5 ions). Quadrupole isolation was set to 0.7 m/z and first mass was set to 50 m/z.

**Data analysis for phosphopeptide library experiments**. Raw MS data were processed with IsobarQuant[59] and peptide and protein identification was performed with the Mascot 2.4 (Matrix Science) search engine. Data were searched against a library containing the respective peptide sequences of the Nterm libraries or Cterm library and the reversed peptides sequences. Each peptide sequence was presented as a separate protein. Search parameters: None, missed cleavages 6, peptide tolerance 10 ppm, 0.05 Da for MS/MS tolerance. No fixed modifications were selected. Oxidation on Methionine and phosphorylation for Serine, Threonine, and Tyrosine were selected as variable modifications.

**Downstream data analysis phosphopeptide library experiments**. After filtering the post-Mascot output files of IsobarQuant, the resulting data files were processed using R (v3.5.1, 2018-07-02) Data were first assessed for empirically wrong peptides (length unequal to ten amino acids, false sequence, no phosphorylation, wrong phosphorylation) and empirically correct peptides (EmpCorr). We then calculated the FDR as the ratio of false positives to the sum of true and false positives for each (sub-) library. For further analysis, we excluded all known empirically wrong peptides and applied the determined cut-off at an FDR of 0.05 to true positive hits (EmpCorr) only. Therefore only a wrongly assigned peptide, which still matched the sequence of another theoretical sequence, could be retained. This step decreased the true percentage of false positives in our final datasets significantly below an FDR of 0.05.

After filtering, we obtained, ranging between conditions, 111,897-147,241 peptides for the Nterm libraries representing 9072–9513 unique sequences (from 19,710 theoretically possible sequences). For the Cterm library we obtained 2065–2362 unique sequences (from 5488 theoretically possible sequences) in sets of 25,066-30,700 peptides (Supplementary Table 3).

The grand average of hydropathy scores (GRAVY scores[60]) were calculated using the free gravy calculator (www.gravy-calculator.de, accessed March 2019). The values of the heat map were calculated for each amino acid in every position independently. The values result from the dephosphorylation rate of an amino acid in a certain position over the average dephosphorylation of the library by the respective phosphatase. The dataset was therefore normalized for variation in amino acid incorporation. The heat map directly comparing PP1c and PP2Ac on the peptide level was calculated by analyzing dephosphorylated peptides of PP1c and PP2Ac and calculating fold changes based on the counts for every amino acid in each position. Heat-maps for Nterm and Cterm libraries were median-normalized to control for differences in global dephosphorylation rates between PP1c and PP2Ac. Statistical significance in Fig. 2 was assessed by applying Fisher's exact test to the numbers of each amino acid at each position in dephosphorylated sequences against the summed counts of all other amino acids at the given position obtained for PP1c versus PP2Ac and adjusting derived p-values for multiple testing using the method of Benjamini and Hochberg.

**Selection of single peptides**. The datasets of Nterm or Cterm libraries treated with recombinant PP1c and PP2Ac were used. The datasets were split into subsets with the following conditions. The pre-filtered datasets were then manually scanned to find the best matching peptide pair, containing small, adequate changes resulting in different dephosphorylation ratios. Syntheses of these peptides and determination of dephosphorylation kinetics by the EnzCheck Assay were performed as described above. Data analysis was carried out in GraphPad Prism v6 and the $k_{cat}/K_m$ was calculated by comparison to a phosphate standard. Error bars in Fig. 2d, e and Supplementary Fig. 3c correspond to s.e.m of independent triplicates with two technical data points each.

**Phosphoproteome dephosphorylation assay**. For phosphoproteomics, HeLa Kyoto cells were incubated with 20 nM Calyculin A (CST, #9902) for 10–15 min. This time point was determined empirically to yield maximal phosphatase inhibition without cells showing pleiotropic effects. Subsequently, cells were placed on ice and washed twice with cold PBS and scraped in lysis buffer (100 mM NaCl, 10 mM Tris, 0.1% IGEPAL, 1 mM EDTA, 1 mM EGTA, 20 nM Calyculin A, 1x cOmplete Mini protease inhibitor cocktail (Sigma), 1 mM DTT, 1 mM $MnCl_2$, pH 7.5). After 10 pushes through an injection needle (21 G, BD), lysate was cleared from the insoluble fraction by centrifugation (10 min, $10^4$ rcf, 4 °C). Protein

concentration was then determined by a Bradford Assay and equal amounts of 1 mg total protein were incubated with recombinant PP1c/PP2Ac or buffer at a final phosphatase concentration of 1 μM in a total volume of 200 μL for 1 h at 30 °C. Lysates were then immediately placed on ice. PhosStop protease inhibitor (Roche) was added and proteins were denatured by adding urea to a final concentration of 8 M taking into account the volume increase. Samples were then snap-frozen in liquid nitrogen for storage and subsequent MS analysis.

Treated lysates were reduced (10 mM DTT, 30 °C, 30 min) and alkylated (50 mM chloroacetamide, room temperature, 30 min, in the dark), and then diluted to 1.3 M urea using 40 mM Tris-HCl (pH 7.6). Digestion was performed by adding trypsin (Promega, 1:50 enzyme-to-substrate ratio) and incubating 3 h at 37 °C at 700 rpm and overnight at 37 °C at 700 rpm upon second trypsin addition. Digests were acidified by addition of neat FA to 1% and desalted using 50 mg tC18, reversed-phase (RP) solid-phase extraction cartridges (Waters Corp.; wash solvent: 0.1% FA in 2% ACN; elution solvent: 0.1% FA in 50% ACN). Peptide solutions were frozen at −80 °C and dried in a SpeedVac (Thermo Fisher).

First, TMT labeling and high pH RP tip fractionation of phosphoproteomes was performed[61]: For TMT labeling, digests (300 μg per condition) were reconstituted in 20 μL of 100 mM HEPES (pH 8.5), and 5 μL of a 11.6 mM TMT stock (Thermo Fisher Scientific) in 100% anhydrous ACN were added to each sample. The different experimental conditions were distributed within a TMT-6plex experiment as follows (each of three replicates multiplexed in an individual TMT6 batch): 126—untreated; 127—PP1 treated; 128—PP2A treated; (129—condition 3, not included in analysis; 130—pooled sample, not included in analysis; 131—empty). After incubation for 1 h at 25 °C and 400 rpm, the labeling reaction was stopped by adding 2 μL of 5% hydroxylamine (15 min, at 25 °C and 400 rpm). Peptide solutions were pooled and acidified using 20 μL of 10% FA. Reaction vessels, in which the labeling took place were rinsed with 20 μL of 10% FA in 10% ACN (5 min, at 25 °C and 400 rpm), and the solvent was added to the pooled sample. The pools were frozen at −80 °C and dried down in a SpeedVac. Subsequently, pooled samples were desalted using 500 mg tC18, RP solid-phase extraction cartridges (Waters Corp.; wash solvent: 0.1% FA in 2% ACN; elution solvent: 0.1% FA in 50% ACN). Peptide solutions were dried in a SpeedVac and frozen at −80 °C.

Phosphopeptides were enriched from the desalted and labeled digests (approximately 1.5 mg) using Fe-IMAC[62]. Fe-IMAC eluate was desalted using C18 StageTips[63], and dried down by vacuum centrifugation on a SpeedVac. High-pH RP tip fractionation was performed in Stage Tips[61]. Tips were washed using 250 μL of 100% ACN, followed by 250 μL of 50% ACN in 25 mM NH₄COOH, pH 10 and then equilibrated with 250 μL of 25 mM NH₄COOH, pH 10. Subsequently, the desalted peptides were reconstituted in 50 μL of 25 mM NH₄COOH, pH 10, and slowly loaded onto the C18 material. After re-application of the flow-through, bound peptides were eluted using 40 μL of solvent with increasing concentrations of ACN (5, 10, 15, 17.5, 50% ACN) in 25 mM NH₄COOH, pH 10. The 5 and 50% ACN fractions were pooled and the 17.5% ACN fraction was combined with the previously stored flow-through, resulting in a total of four fractions, which were dried and stored at −20 °C until LC-MS/MS measurement.

**Mass spectrometric data acquisition for phosphoproteomics.** LC-MS/MS measurements of TMT6-plex-labeled phosphopeptides were carried out using a Dionex Ultimate3000 nano-HPLC coupled online to a Fusion Lumos Tribrid mass spectrometer (Thermo Fisher Scientific). Peptides were dissolved in 15 μL citrate solution (0.1% FA and 50 mM citric acid) and one third was injected. Peptides were delivered to a trap column (75 μm × 2 cm, packed in-house with 5 μm C18 resin; Reprosil PUR AQ, Dr. Maisch, Ammerbruch-Entringen, Germany) and washed using 0.1% FA at a flow rate of 5 μL/min for 10 min. Subsequently, peptides were transferred to an analytical column (75 μm × 45 cm, packed in-house with 3 μm C18 resin; Reprosil Gold, Dr. Maisch) applying a flow rate of 300 nL/min and separated using a 90 min linear gradient from 4% to 32% LC solvent B (0.1% FA, 5% DMSO in ACN) in LC solvent A (0.1% FA in 5% DMSO). The eluate from the analytical column was sprayed with a stainless steel emitter (Thermo Fisher Scientific) at a source voltage of 2.1 kV into the mass spectrometer. The transfer capillary was heated to 275 °C. The Fusion Lumos was operated in positive ionization mode and data-dependent acquisition (DDA), automatically switching between MS1, MS2, and MS3 spectrum acquisition. Full scan MS1 spectra were recorded in the Orbitrap from 360 to 1300 m/z at a resolution of 60k (automatic gain control (AGC) target value of 4e5 charges, maximum injection time (maxIT) of 50 ms). Precursor ions that were singly charged, unassigned, or with charge states >6+ were excluded. Up to 10 peptide precursors were isolated (isolation window, 0.7 m/z; maximum injection time, 22 ms; and AGC value, 5e4), fragmented by collision-induced dissociation (CID) using 35% normalized collision energy (NCE) and multistage activation (−80 Da trigger), and analyzed at a resolution of 15k in the Orbitrap. For subsequent MS3 spectral acquisition, fragment ions were selected by multi-notch isolation, allowing a maximum of 10 notches and a charge-state dependent isolation window (2 + : 1.2 m/z, 3 + : 0.9 m/z, 4 + : 0.7 m/z, 5+ and 6 + : 0.6 m/z). These ions were fragmented by higher-energy collision-induced dissociation (HCD) at 55% NCE (AGC target value 1.2e5, maximal injection time 120 ms), and the resultant MS3-fragment ions were recorded in the Orbitrap with a m/z range of 100–1000 and a resolution of 50k. The dynamic exclusion duration of fragmented precursor ions was 90 s.

**Phosphoproteomic data analysis.** Proteins and peptides were identified and quantified using MaxQuant (v1.6.0.16)[64] with enabled MS3-based TMT quantification using default parameters and searching spectra against all canonical protein sequences as annotated in the Swissprot reference database (human proteins only, 20193 entries, downloaded 22 March 2016, internally annotated with PFAM domains) using the embedded search engine Andromeda[65]. Oxidation of methionine and N-terminal protein acetylation as well as phosphorylation on S, T, and Y and oxidation of M were specified as variable modification. Carbamidomethylation on cysteines was specified as a fixed modification. Trypsin (Trypsin/P) was specified as the proteolytic enzyme with up to two allowed missed cleavage sites. Quantification based on TMT-6plex MS3 reporter ion intensity was enabled, and the results were filtered for a minimal length of seven amino acids and 1% peptide and protein FDR MaxQuant results were imported into the MaxQuant associated software suite Perseus (v.1.5.8.5)[66]. Phosphosites were filtered for at least three valid values for at least one experimental group. TMT batch effects were reduced by row-wise normalization based on median intensities and all further steps of analysis are described in detail in their respective sections.

**Downstream data analysis of phosphoproteomic experiments.** Downstream analysis of the phosphoproteomic datasets was carried out using the software tool Perseus[66]. For analysis of phosphoproteomic experiments, row-wise, median-normalized data (see previous section) was linked to known regulatory sites by PhosphoSitePlus entries and categorical columns for treatments were assigned. Reverse peptides and potential contaminants were filtered. Next, categorical columns for biological functions and treatments were assigned. TMT reporter intensities were then transformed into log2. At this point, all data were then filtered for valid values in all three replicates of at least one condition. Next, the distribution of TMT reporter intensities was assessed visually by plotting count-based histogramms and in a multiscatter plot using the Pearson correlation as an estimate of reproducibility. Next, NA (=not available: missing) values were replaced by imputation from a normal distribution (width 0.3, down-shift 1.8, per column). The data quality was routinely assessed by two means: First, by plotting a histogram of peptide length vs. counts and visualization of imputed values to confirm a normal distribution of the data, and secondly by principal component analysis (default settings in Perseus) to judge the sample grouping.

At this stage, data for Volcano plots were produced by t-tests (default settings: two-sided, grouped by conditions, 250 randomizations, FDR 1%, s0: 0.1) These data as well as curves for significance were then exported and plotted in R using ggplot2. For hierarchical clustering, the results of the one-way ANOVA were processed for two more steps in Perseus: first data were filtered for significant hits from the one-way ANOVA and then subjected to z-transformation. Hierarchical clustering images were produced using Euclidean distance, 300 clusters and 10 rounds of iteration.

Also all further data processing for protein enrichment and extraction of protein sequences in positions −5 to +5 relative to the p-sites of interest for subsequent motif analysis was performed using RStudio with R v3.3.2 using the packages tidyverse and BioStrings. In order to study the sequence specificity of PP1c and PP2Ac on the protein level, position specific probability matrices, representing the relative occurrence for each amino acid surrounding pSer/pThr were produced by extracting sequence information in positions −5 to +5 for all PP1c/PP2Ac-sensitive class I p-sites. Next, the count for each amino acid in each position was divided by the total of all counts for a certain position, resulting in relative abundance. The fold-change for depiction in heat-maps was calculated by dividing the relative abundances between pSer/pThr class sites significantly dephosphorylated upon phosphatase treatment compared to p-sites without significant changes. Importantly, amino acids with less than 25 counts in PP1-sensitive/-insensitive or PP2A-sensitive/insensitive comparisons were excluded. For calculating a heat map directly comparing PP1c against PP2Ac, these fold changes were divided. Statistical significance in Figs. 3d, 4b was assessed by applying Fisher's exact test to each amino acid in each position between significantly dephosphorylated and unchanged peptide sequences for PP1 versus PP2A incubated samples. Obtained p-values were adjusted for multiple testing using the method of Benjamini and Hochberg.

**Gel-filtration analysis of the phosphoproteomic assay.** In order to control for a potential formation of holoenzymes of recombinant PP1c and PP2Ac upon incubation with HeLa Kyoto cell lysate, the dephosphorylation assay presented above was also analyzed by gel filtration, with the only difference of using His-PP1c for being able to discriminate between recombinant and endogenous PP1c. After incubation, the assay was injected into a 200 μL sample loop and loaded onto a Superdex 200 Increase 10/300 column installed on an ÄKTA explorer system (both GE Life Sciences) at room temperature and with a flow rate of 0.25 mL/min and using 10 mM Tris, 100 mM NaCl, 0.1% IGEPAL (pH 7.5) as equilibration and elution buffer. Fractions were automatically collected in 96-well plates. 30 μL of 500μL-fractions corresponding to elution volumes of molecular weights between approx. 30-700 kDa were analyzed by gel electrophoresis. Subsequently, gels were analyzed by colloidal Coomassie staining or Western blotting. Since the UV-absorbing properties of IGEPAL in the lysis buffer prevented the detection of elution peaks at 280 nm, the gel-filtration standard mix of six proteins (#MWGF1000, Sigma-Aldrich) was compared in buffer with and without 0.1%

IGEPAL. For the positive control, 1 μM His-PP1c and and Inhibitor-2 (I-2, IPP2, PPP1R2) were incubated in 200 μL assay buffer for 1 h before gel filtration.

**Analysis of molecular weights of PP1 and PP2A regulatory subunits.** Manually curated sets of all annotated PP1 and PP2A regulator subunits were downloaded from the website of the HUGO gene nomenclature committee (HGNC, www.genenames.org, accessed Feb 2020). Subsequently, HGNC IDs were correlated to Uniprot IDs using the uniprot online tool and information about the molecular weight of the canonical isoform for all regulatory subunits was downloaded in batch mode (www.uniprot.org, accessed Feb 2020). Data were then imported and further analyzed in R.

**PSSMSearch motif analysis.** PP1c/PP2Ac-sensitive class I sites were divided into three classes as illustrated for PP1c in Supplementary Fig. 4c. All PP1c/PP2Ac-insensitive sites (not siginifcant) were used as controls. The relative amino acid distributions of residues −5 to +5 for the indicated subsets were then calculated by extracting the sequence surrounding pSer/pThr. These were then uploaded to PSSMSearch (http://slim.ucd.ie/pssmsearch/, accessed between July and December 2018) in batch mode. Sequences were then analyzed by using the unbiased scoring method'frequency'. The statistically significant enrichment of Arg in PP1c-sensitive sites compared to PP2Ac-sensitive sites depicted in Fig. 5a was calculated using the two-sided Fisher's exact test in R.

**Co-Immunoprecipitation (Co-IP) experiments for MS analysis.** HeLa Kyoto cells expressing transiently transfected pEGFP(C1)-14-3-3β or pEGFP(C1) control plasmid, were incubated with 20 nM Calyculin A (# 9902 S, Cell Signaling Technology) for 12 min. Cells were washed twice on ice with cold PBS and after addition of 500 μL lysis buffer (100 mM NaCl, 10 mM Tris, 0.1% IGEPAL, 1 mM EDTA, 1 mM EGTA, 20 nM Calyculin A, 1x cOmplete Mini protease inhibitor cocktail (Sigma), 1 mM DTT, 1 mM MnCl$_2$) cells were scraped from plates, lysed by 10 pushes through an injection needle (21 G, BD) and the insoluble fraction was pelleted by centrifugation (10 min, $10^4$ rcf, 4 °C). Supernatants were then subjected to incubation with 1 μM recombinant phosphatase or buffer for 1 h at 30 °C. Next, we subjected samples to immunoprecipitation using α-GFP nanobodies covalently coupled to agarose beads (2 h, 4 °C, obtained from the EMBL PEPcore Core Facility). Beads were then washed three times with lysis buffer and samples were eluted in 1x LDS sample buffer (Thermo Fisher Scientific) by heating to 70 °C for 10 min.

**Sample preparation and LC-MS/MS of 14-3-3 co-IP.** The samples were then reduced with 25 mM dithiothreitol (10 min at 70 °C) and alkylated with 55 mM chloroacetamide (room temperature, 30 min, in the dark). Proteins were run on a precast 4–12% NuPAGE gel (ThermoFisher Scientific) for about 1 cm to concentrate the sample prior to in-gel tryptic digestion, which was performed according to the standard procedures[67]. The peptides obtained were dried to completeness and resuspended in 12 μL of buffer A (0.1% FA), and 5 μL of sample were injected per MS measurement. The experiment was carried out in three independent replicates.

The samples were analyzed by LC-MS/MS on a Dionex Ultimate 3000 nano-HPLC coupled online to a Fusion Lumos Tribrid mass spectrometer (Thermo Fisher Scientific). Peptides were delivered to a trap column (75 μm × 2 cm, packed in-house with 5 μm C18 resin; Reprosil PUR AQ, Dr. Maisch) and washed using 0.1% FA at a flow rate of 5 μL/min for 10 min. Subsequently, peptides were transferred to an analytical column (75 μm × 45 cm, packed in-house with 3 μm C18 resin; Reprosil Gold) applying a flow rate of 300 nL/min and separated using a 60 min linear gradient from 4% to 32% LC solvent B (0.1% FA, 5% DMSO in ACN) in LC solvent A (0.1% FA in 5% DMSO). The eluate from the analytical column was sprayed via a stainless steel emitter (Thermo Fisher Scientific) at a source voltage of 2.1 kV into the mass spectrometer. The transfer capillary was heated to 275 °C. The Fusion Lumos was operated in positive ionization mode and data-dependent acquisition (DDA), automatically switching between MS1 and MS2 spectrum acquisition. Full scan MS1 spectra were recorded in the Orbitrap from 360 to 1300 $m/z$ at a resolution of 60k (automatic gain control (AGC) target value of 4e5 charges, maximum injection time (maxIT) of 50 ms). Up to 20 peptide precursors were isolated (isolation window, 1.7 $m/z$; maximum injection time, 25 ms; and AGC value, 1e5), fragmented by high-energy collision-induced dissociation (HCD) using 30% normalized collision energy (NCE), and analyzed at a resolution of 15k in the Orbitrap. Precursor ions that were singly charged, unassigned, or with charge states >6+ were excluded. The dynamic exclusion duration of fragmented precursor ions was 20 s.

Peptide and protein identification and LFQ were performed using MaxQuant software (version 1.6.1.0)[64] by searching the data against all canonical protein sequences as annotated in the Swissprot reference database (human proteins only, 20193 entries, downloaded 22 March 2016, internally annotated with PFAM domains) using the embedded search engine Andromeda[65]. The MaxQuant search was performed using two variable modifications; oxidation of methionine and N-terminal protein acetylation. Carbamidomethylation on cysteines was specified as a fixed modification. Trypsin (Trypsin/P) was specified as the proteolytic enzyme with up to two allowed missed cleavage sites. LFQ[68] and match-between-runs

(matching time window of 0.7 min and alignment time window of 20 min) were enabled, and the results were filtered for a minimal length of seven amino acids and 1% peptide and protein FDR. MaxQuant results were imported into the MaxQuant associated software suite Perseus (v.1.5.8.5)[66] and further analyzed as laid out in detail in the following section.

**Downstream data analysis of MS-Co-IP dataset.** First, reverse peptides and potential contaminants were filtered. Next, categorical columns for biological functions and treatments were assigned. LFQ values were then transformed into log2. At this point, all data were then filtered for valid values in all three replicates of at least one condition. Next, distribution of LFQ intensities was assessed visually by plotting count-based histograms and in a multiscatter plot using the Pearson correlation as an estimate of reproducibility. Next, NA values were replaced by imputation from a normal distribution (width 0.3, down-shift 1.8, per column). Data quality was again assessed by plotting a histogram highlighting the population of imputed values and by principal component analysis using default settings.

At this stage, data for the Volcano plots were produced by t-tests (default settings: two-sided, grouped by conditions, 250 randomizations, FDR 5%, s0: 0.1). Only proteins with three hits in at least one condition entered analysis. These data as well as curves for significance were then exported and plotted in R with ggplot2.

The overlap between phosphoproteomic experiments and MS-Co-IP data was analyzed by extracting all significantly dephosphorylated class I sites (localization probability >0.75) between untreated and PP1-sensitive p-sites and comparing Uniprot IDs with 56 proteins that significantly dissociated from eGFP-14-3-3β upon treatment with PP1 but did not bind eGFP alone.

**In vitro dephosphorylation of GAB2 and 14-3-3 binding assay.** pEGFP(C1)-14-3-3β or pCMV-GAB2-3FLAG was transfected into HeLa Kyoto cells as described in the respective section for 24 h. Cells were incubated with or w/o 20 nM Calyculin A (# 9902 S, Cell Signaling Technology) in complete growth media for 12 min. Cells were washed twice on ice with cold PBS and after addition of 500 μL lysis buffer (100 mM NaCl, 10 mM Tris, 0.1% IGEPAL, 1 mM EDTA, 1 mM EGTA, 20 nM Calyculin A, 1x cOmplete Mini protease inhibitor cocktail (Sigma), 2 mM DTT, 2 mM MnCl$_2$) cells were scraped from plates, lysed by 10 pushes through an injection needle and the insoluble fraction was pelleted by centrifugation (10 min, $10^4$ rcf, 4 °C). Supernatants were then subjected to incubation with 1 μM recombinant PP1c (if not indicated otherwise) or buffer for 1 h at 30 °C. For pre-inhibition of recombinant PP1c, equimolar amounts of PP1CA and Calyculin A were mixed for 5–10 min prior to addition to the lysate. Next, we subjected samples to immunoprecipitation/pull-down using 25 μL of α-FLAG M2 magnetic bead slurry (Sigma-Aldrich) or an equal volume of bead-coupled α-GFP nanobodies (EMBL PEPcore Core Facility) for 2 h at 4 °C. Beads were then washed three times with lysis buffer and samples were eluted in 1x SDS sample buffer by heating to 95 °C for 10 min and further analyzed by SDS PAGE for 14-3-3/GAB2 association and phosphorylation status of GAB2-pS210.

**Protein gel electrophoresis and immunoblotting.** For SDS-PAGE, samples were loaded on NuPAGE™ 4-12% Bis-Tris protein gels (Invitrogen) and gel electrophoresis was performed at 180 V constant in 1x MOPS buffer. Samples were transferred onto CN blotting membranes (0.45 μm, neoLab) by wet blotting. Membranes were stained using Ponceau, followed by blocking with Tris-buffered saline with 0.1% Tween (TBS-T) containing 5% (w/v) non-fat milk for 45 min at room temperature. After three washes (5 min), membranes were then incubated with the following antibodies in TBS-T with 5% BSA overnight at 4 °C: α-GFP (1:2000, Abcam, ab6556), α-GAB2 (1:1000, CST, #3239), α-GAB2-pS210 (1:250, Symansis, customized purification), α-FLAG (1:1000, M2, Sigma-Aldrich, F3165), α-pan14-3-3 (1:500, SCBT, sc-629), α-PP1 (1:1000, SCBT, sc-7482), α-PP2A (1:1000, SCBT, sc-130237) or α-6His (1:2000, Abcam, ab9108). Afterwards blots were washed three times in TBS-T and incubated with goat anti-mouse IgG-peroxidase conjugate (1:2500, Sigma, A0168), donkey anti-rabbit IgG-horseradish peroxidase (HRP) conjugate (1:2500, GE Healthcare, GENA934), or rabbit anti-sheep IgG (H + L) HRP conjugate (1:2500, Thermo Fisher Scientific, 61-8620) for 2 h at room temperature. After three final washes with TBS-T, blots were developed using ECL Western Blotting Detection Reagents (GE) and Western Lightning Plus-ECL Enhanced Chemiluminescence Substrate (Perkin Elmer) and using a ChemiDoc Touch Imaging System (BioRad) or a Fusion FX Imaging System (Vilber). All blots were quantified using raw images and ImageLab software (v6.0.1, BioRad): Files were imported and analyzed without further modifications. Lane boundaries were set manually. The band detection was carried out using the auto-analysis tool. After band detection, the band profiles were inspected and the boundaries were adjusted if necessary. The quantification was exported from the analysis table and measurements of background adjusted band volumes where used for further normalization steps and graphical depiction. The statistic shown in Fig. 6a, b was obtained by performing a paired, two-sided Student's $t$ test and subsequent adjustment of p-values for multiple testing.

**Directed peptide synthesis of PDP(m)-Nal.** PDP-Nal and PDPm-Nal were synthesized using a standard fluorenylmethoxycarbonyl (Fmoc)/solid-phase peptide

synthesis protocol[36]: First, Fmoc-amino acids (5 eq) were linked by double couplings using diisopropylethylamine (DIEA, 6 eq), HBTU, HOBt (both5 eq), DMF and Rink amide resin. Capping and N-terminal acetylation was carried out using a ratio of 1:9 (vol:vol) of acetic anhydride:pyridine. 20% Piperidine was used for Fmoc deprotection and 95% TFA, 2.5% TIPS, 2.5% $H_2O$ for cleavage from the resin. Subsequently, peptides were ether-precipitated, purified by HPLC, characterized by MS and dried for storage. For experiments all peptides were used from a 10 mM stock in 5% DMSO in water.

**Confocal microscopy**. The cells were seeded on 8-well ibiTreat µslides (Ibidi) and transfected as described in the respective section. In order to serum-starve cells for 45 min to 1 h before imaging, cells were washed twice with warm PBS and cultivated in 200 µL DMEM without FBS and phenol red (Gibco) supplemented with 1 g/L L-glutamine and 1% penicillin/streptomycin. For imaging, cells were then mounted in an Okolab stage incubator system (37 °C, 5% $CO_2$ using an H101-CRYO-BL temperature controller, $CO_2$-$O_2$ UNIT-BL gas controller, H101-HM active humidity controller and incubation chamber H101-NIKON-TI-S-ER). All images were acquired with the A1R Confocal Scanning System equipped on a Nikon Ti-E inverted microscope installed with an Apo TIRF 100x DIC N2 1.49 NA Oil objective. A motorized stage and the Nikon perfect focus system were used to guarantee stable conditions during the experiments. mKate2 was excited with the 561 nm line from a 50 mW Sapphire solid-state laser from Coherent, selected with an AOTF and collected with a 405/488/561 dichroic mirror (Chroma) and a bandwidth filter of 595/50 nm. Images were acquired using the GaAsP detectors controlled with the NIS Elements 4.50 operating software. Time-lapse-experiments consisted of an image acquisition interval of 15 s for at least 12 min for each sample condition separately using an exposure time of 0.5 frames per second. Cells were treated after frame 3 by adding 100 µL of starvation media (see above) with 300 ng/mL EGF (Gibco) or 150 µM PDP(m)-*Nal* resulting in final concentrations of 100 ng/mL for EGF and 50 µM for PDPs.

**Image analysis and quantification**. For image analysis, microscopy files were loaded into Fiji v2.0.0-rc-34/1.50a. First, images at 0 min and 5 min (HeLa Kyoto) or 12 min (Caco-2 BBe1, SW-480) were extracted, followed by background subtraction. To this end, the threshold was taken to the upper limit to select an area without cells, but the threshold was not applied on the image. Within this area, a measurement for the mean gray value was taken and subtracted from the entire image using [Process-Math-Subtract]. The threshold was then adjusted to the cell boundaries and used to transform the image into a binary image. Next, segmentation was carried out using a median filter (2px), and the functions 'fill holes' and 'watershed'. For manual correction of cell segmentation, the original image was used as an overlay. The result was saved as mask 1. Next, an erosion step with a minimum filter was applied using [Process-Filter-Minimum (15 px)]. The result was saved. Using [Process-ImageCalculator], the latter image was then subtracted from mask 1. The result was saved as mask 2. Masks were then applied onto the raw images at 0 min and 5 or 12 min, respectively, and particle analysis was used for determining the mean gray value within the selected areas of mask 1 and 2 on the single cell level. These values were then further processed as shown for the analysis of HeLa Kyoto cells (5 min) in Eq. (1):

$$f_{GAB2} = \frac{m_{mask2(5min)}/m_{mask1(5min)}}{m_{mask2(0min)}/m_{mask1(0min)}}, \tag{1}$$

where $f_{GAB2}$ is a numerical factor for GAB2 accumulation at the cell periphery, $m$ is the mean gray value, mask 1 is nucleus and cytoplasm and mask 2 is cell periphery and membrane. This resulted in the fold-enrichment of GAB2 at the plasma membrane for EGF/PDP(m)-*Nal* addition.

These data were then imported into R/RStudio and plotted using ggplot2. Wilcoxon signed-rank test was carried out to assess significance in Fig. 6e and Supplementary Fig. 8. Obtained p-values were adjusted for multiple testing with the Benjamini–Hochberg method.

**Statistics**. Student's t test, Wilcoxon test, Fisher's exact test (95% confidence interval) and one-way ANOVA have been used in the course of this study. For clarity we have described the specific test details at the end of each respective experimental sections. In case adjusted p-values are reported in order to control for multiple comparisons, Benjamini–Hochberg correction was applied. The underlying results and exact p-values of statistical analysis using the Fisher's exact test are also shown in the Source Data.

**Reporting summary**. Further information on research design is available in the Nature Research Reporting Summary linked to this article.

## Data availability

Mass spectrometry data have been deposited at the ProteomeXchange Consortium (proteomecentral.proteomexchange.org) via the PRIDE partner repository with the identifiers PXD012026[http://proteomecentral.proteomexchange.org/cgi/GetDataset?ID=PXD012026] and PXD013775[http://proteomecentral.proteomexchange.org/cgi/GetDataset?ID=PXD013775]. The output of all MS-based results is furthermore

summarized in Supplementary Tables 1–3 and Supplementary Data 1–3. Supplementary Tables 1–3 as well as Supplementary Figs. 1–10 are found in the Supplementary Information, Supplementary Data 1–3 are provided as excel files. Source data are provided with this paper. All other data supporting the findings presented herein are available from the authors upon request.

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

## Acknowledgements

The authors thank Pavel Salavei (Signalling Factory Uni Freiburg) for the help with protein and library purification, Nicole Gensch (Signalling Factory, BIOSS and CIBSS Uni Freiburg) for assistance with the live-cell microscopy setup, Maximilian Hörner (Faculty of Biology, BIOSS and CIBSS Uni Freiburg) for assistance with the GAB2 antibodies, Tilman Brummer and Sebastian Halbach (IMMZ Uni Freiburg) for the GAB2 antibodies, plasmids and discussions, Lena Reimann and Bettina Warscheid (Faculty of Biology, BIOSS and CIBSS Uni Freiburg) for advice in MS analysis, the EMBL PEPCore for preparation of PP1c and bead-coupled GFP nanobodies, and Christina Gross (CIBSS Uni Freiburg) for scientific and language editing of the manuscript.This work was supported by an ERC Starting Grant (#336567) and an ERC Consolidator Grant (#865119) to M.K., and by funding from the Deutsche Forschungsgemeinschaft DFG through EXC 294 BIOSS and SFB 850 (M.K.).

## Author contributions

B.H. designed study parts, optimized and carried out experiments, established and carried out data analysis, and wrote the manuscript. T.K. carried out peptide library dephosphorylation assays and analyzed the data. J.C. synthesized and optimized peptide libraries. D.H. co-designed the PLDMS approach, performed LC-MS/MS measurements of peptide library samples and processed primary MS data. S.H. and C.L. designed and carried out LC-MS/MS measurements for proteomics experiments and processed primary MS data. T.S. optimized and performed live-cell microscopy experiments. A.B. and B.H. carried out the computational structural analysis. N.K. carried out statistical analysis. A.B. T.K., D.H., M.M.S., T.S., C.L., S.H., and B.K. edited the manuscript. M.M.S. supervised phosphopeptide library MS-experiments. B.K. supervised phosphoproteomic experiments. M.K. designed, arranged, and supervised the study, and wrote and edited the manuscript.

## Competing interests

The authors declare no competing interests.
