## [Peer Review File · Nature Communications]

Reviewers' comments:

Reviewer #1 (Remarks to the Author):

PP1 and PP2A are major cellular S/T phosphatases and understanding of their substrate selectivities is a fundamentally important question relevant to all biomedical fields. The study by Köhn laboratory addresses this question by phosphoproteome analysis of both peptide libraries and cellular lysates treated transiently with recombinant PP1 and PP2A catalytic subunits (PP1c and PP2Ac, respectively).

Although some of the main conclusions have been suggested by the previous studies, this work provides very important unbiased information to complement our understanding of PP1 and PP2A intrinsic substrate preferences. Authors have performed extensive quality control experiments to minimize the likelihood that the observed results are not due to technical bias. However, many results require statistical significance analysis which would be essential to evaluate the significance of the findings, but could also help the authors to further squeeze out from their data the most reliable and meaningful conclusions.

General comments:

1. As this study potentially fills previous knowledge gaps and substantiates the previous findings related to substrate peptide selectivity of PP1c and PP2Ac, it would be very important to clearly discuss what is the advance from this study, as compared to previous studies, and what would be the current master model when putting all existing studies together. In addition to discussing in more detail the model in light of their own data and already referenced papers from Saurin, Nilsson and Peti laboratories, they should also include other recent important studies such as the study by Hendus-Altenburger et al., which addressed the importance of flanking aminoacids for Calcineurin-mediated dephosphorylation (Nat Commun. 2019 Aug 2;10(1):3489).

2. It would be very important that authors even more clearly indicate, starting from title, abstract and introduction, that at least currently there are no examples of dephosphorylation of S/T by catalytic subunits alone. Rather the presented results presented are relevant for understanding which S/T residues could be preferentially dephosphorylated by PP1 and PP2A, AFTER they have been recruited to the target/target complex by regulatory subunits. Thereby the authors could prevent misunderstanding that they are proposing a new mode of function for PP1c and PP2Ac as

monomers. Along these lines, starting from the title, I would not talk about PP1 and PP2A substrate specificity but rather about amino acid sequence determinants for dephosphorylation by catalytic subunits of PP1 and PP2A

3. Please explain clearly in the text that even though catalytic subunits have T/S preference, still vast majority of all dephosphorylated sites in our cells are serines. Otherwise the reader might leave with wrong perception that majority of dephosphorylation events by these PPases occur on Threonines.

Comments to data and figures

4. To ensure the purity of proteins, please provide full gel coomassie staining images of purified PP1c and PP2Ac used in the experiments.

5. Figure 1: What is the basis for using AAAA stretches before and after the phosphorylated S/T? How is it controlled that this not affect the results as no such motifs are found in nature in most target sites.

6. Fig. 2B: The patterns for PP1 and PP2A look pretty similar. Which amino acid preferences are statistically significantly different between PP1 and PP2A?

7. Fig. 3A):

-Doesn't CalA in the lysate inhibit also the recombinant PP1c and PP2Ac?

- What is the approximate molar ratio between recombinant PPase and substrate proteins in the lysate

8. In. 240: This is a very important potential caveat of the study. Please demonstrate the degree of incorporation of recombinant PP1c and PP2Ac to holoenzyme complexes by gel-filtration analysis and affinity purification coupled with MS.

9. 3D) Are these enrichments statistically significant?

10. 3E) Similar to 2B, which aminoacid preferences are statistically significantly different between PP1 and PP2A? Even more importantly, after calculating the significances for both 2B and 3E, what are the aminoacid statistically significant aminoacid preferences that are consistent between both methods (Fig. 2 and 3). This would be very important and clear take a home message from the paper with statistical power.

11. Fig. 4: It is unclear whether this analysis is based on data from Fig. 2 or Fig. 3? Again, related to comments above, similar analysis should be done with both datasets.

12. Ln. 276. Is the authors final conclusion that after all there is no aminoacid preference for PP2Ac?? This is very confusing as related to overall message of the paper. Please clarify.

13. Ln. 281-283: The alternative explanation would be that as PP1 functions a s dimer it more easily assembles to holoenzyme than PP2A in these in vitro lysate conditions and therefore its shows more strict substrate selectivity. This further indicates that authors need to provide the experimental evidence requested in my comment 8.

14. Fig. 4C; In. 286-291: Are these enrichments statistcially significant between PP1 and PP2A.

15. Ln. 38: I would prefer that the authors would not strengthen the misconception of the inbalance of kinase and phosphatases. Human genome has 40 genes that code for the catalytic subunit of S/T PPases, but at least as many functional PPases (as functional complexes) as there are kinases.

16. Ln. 180-181: "that the close proximity of Pro to the pSer/Thr could have suggested an active-site-mediated recognition of the SP motif" For a non-expert reader this is hard to understand

17. In. 259: "class I sites (localization probability of phosphorylation >0.75" For a non-expert reader this is hard to understand

Reviewer #2 (Remarks to the Author):

The authors in the manuscript entitled “Dissecting the basic substrate specificity of phosphoprotein phosphatases 1 and 2A through phosphoproteomic approaches” have explored large phosphopeptide library to show the pThr specificity of PP1 and PP2 phosphatases. The generation and validation of a reliable phosphopeptide library is crucial for this study. Therefore, authors have purified and verified great deal of different phosphopeptides to carry out PP1 and PP2A dephosphorylation assays. The biggest flaw effecting the whole study is the fact that only recombinant catalytic subunits (PPP1CA & PPP2CA) were used in the dephosphorylation assays. PP1 and PP2A, as well as other protein phosphatases form large protein complexes with multiple regulatory subunits regulating their activities (for example PMID: 27880917 and PMID: 28330616). The main open question is how and if the substrate specificity/preference changes when the PP1 and PP2 regulatory subunit(s) are also used in the assays?

Nevertheless, the authors also tested their conclusion by in vivo (HeLa cell line) phosphoproteomics analysis, where both PP1 and PP2 again showed intrinsic preference for pThr sites. They have also investigated if this preference is somehow affected by the any presence of other amino acids around the PThr/pSer. In this quest they have found RXXpS motif, in several PP1 and PP2 substrates as well as in 14-3-3 binding site. One of the identified substrates is GAB2 which also bind with 14-3-3 binding site containing proteins (14-3-3 beta) have been dephosphorylated by PP1 in vitro, leading to disruption of GAB2-14-3-3 interaction. This extent of dephosphorylation was also monitored by MS using label-free quantification, where they found PP1 specificity towards RXXpS motif.

Major criticism:

The authors should verify how the presence of the regulatory subunit(s) affect PP1 and PP2A substrate specificity.

Minor comments:

Only one cell line was used in vivo assay. The use of (an)other cell line(s) would strengthen the manuscript. Additionally for authentication of the cell line statement “to the best of our knowledge these are HeLa cells” is not sufficient. See for example a great study on the HeLa heterogeneity by the Aebersold group “PMID: 30778230”

Minor corrections

-Page 19 , line 10 ----(Fig 6a) should be Fig 5c,

line 11-----Fig 6b should be Fig 5d,

line 12----- (Fig 6c) should be Fig 5e,

line 14----- (Fig 5b) should be Fig 5c.

-Figure 5a should show error bars. In its legends, e, should be legend d and f, should be legend e.

Answers to the reviewers' comments:

Reviewer #1 (Remarks to the Author):

PP1 and PP2A are major cellular S/T phosphatases and understanding of their substrate selectivities is a fundamentally important question relevant to all biomedical fields. The study by Köhn laboratory addresses this question by phosphoproteome analysis of both peptide libraries and cellular lysates treated transiently with recombinant PP1 and PP2A catalytic subunits (PP1c and PP2Ac, respectively).

Although some of the main conclusions have been suggested by the previous studies, this work provides very important unbiased information to complement our understanding of PP1 and PP2A intrinsic substrate preferences. Authors have performed extensive quality control experiments to minimize the likelihood that the observed results are not due to technical bias. However, many results require statistical significance analysis which would be essential to evaluate the significance of the findings, but could also help the authors to further squeeze out from their data the most reliable and meaningful conclusions.

Authors: We would like to thank the reviewer very much for the thorough feedback. We have now refined our statistical analysis and added the results to all points raised below. Briefly, in the revised manuscript the statistics now indeed enable us to highlight the power of the library-based approach due to the high number of data points. In addition, we are now also able to derive the proposed conclusions for an intrinsic basophilic amino acid preference of PP1 and pT preference for both phosphatases with statistical power.

General comments:

1. As this study potentially fills previous knowledge gaps and substantiate the previous findings related to substrate peptide selectivity of PP1c and PP2Ac, it would be very important to clearly discuss what is the advance from this study, as compared to previous studies, and what would be the current master model when putting all existing studies together. In addition to discussing in more detail the model in light of their own data and already referenced papers from Saurin, Nilsson and Peti laboratories, they should also include other recent important studies such as the study by Hendus-Altenburger et al., which addressed the importance of flanking aminoacids for Calcineurin-mediated dephosphorylation (Nat Commun. 2019 Aug 2;10(1):3489).

Authors: So far, despite evidence for amino acid sequence determinants for the PP1/PP2A substrates and their biological relevance, the underlying reasons were poorly understood. We designed our approaches to test whether the catalytic subunits of PP1 and PP2A are primed towards certain amino acids around phosphorylation sites and to enable separating this potential basic layer from the contribution of regulatory subunits based on previous studies. Therefore, one advance is the description of the contribution of the substrate specificity of the catalytic subunits to the whole regulation of specificity of the two phosphatases through holoenzymes. We thereby focused here on the basophilic motif and the pThr preference. Another advance concerns the methodology applied here. Our approaches, in particular the PLDMS approach, are novel and thanks to the reviewers comments on statistics we can now clearly show the advantages of these approaches. We have added sections to highlight the advances in the main text and now also graphically present the biological take-home message of our study in Fig. 8d,e. We also thank the reviewer for the advice to extend the scope of our results to PP2B/Calcineurin and have included the suggested reference (lines 435-441).

2. It would be very important that authors even more clearly indicate, starting from title, abstract and introduction, that at least currently there are no examples of dephosphorylation of S/T by catalytic subunits alone. Rather the presented results presented are relevant for understanding which S/T residues could be preferentially dephosphorylated by PP1 and PP2A, AFTER they have been recruited to the target/target complex by regulatory subunits. Thereby the authors could prevent misunderstanding that they are proposing a new mode of

function for PP1c and PP2Ac as monomers. Along these lines, starting from the title, I would not talk about PP1 and PP2A substrate specificity but rather about amino acid sequence determinants for dephosphorylation by catalytic subunits of PP1 and PP2A.

Authors: As the reviewer said, we are at no point arguing against the established holoenzyme-based model for PP1/PP2A functionality. We fully agree that this message needs to be communicated as clearly as possible to avoid any misconceptions, which as the response of the second reviewer showed we did not do in our initial manuscript. Our manuscript is about biochemical investigations of an often observed (PMID 23674824), biologically relevant (PMID 31494926), but poorly defined additional, basic layer of phosphorylation site preference of PP1/PP2A. We have changed the wording in the title and abstract accordingly and added multiple statements in the main text to convey this message.

3. Please explain clearly in the text that even though catalytic subunits have T/S preference, still vast majority of all dephosphorylated sites in our cells are serines. Otherwise the reader might leave with wrong perception that majority of dephosphorylation events by these PPases occur on Threonines.

Authors: We would like to thank the reviewer for highlighting this important statement. Indeed, it is a well-established fact that pS is found more often in mammalian cells. We also observed this and corrected the manuscript in order to state this explicitly (lines 208-210).

Comments to data and figures

4. To ensure the purity of proteins, please provide full gel coomassie staining images of purified PP1c and PP2Ac used in the experiments.

Authors: We have now included the requested coomassie staining images of the same protein batches used for all experiments in this manuscript. As can be seen in Supplementary Figure 1a, purity of both catalytic subunits is demonstrated.

5. Figure 1: What is the basis for using AAAA stretches before and after the phosphorylated S/T? How is it controlled that this not affect the results as no such motifs are found in nature in most target sites.

Authors: In order to not only simulate phosphorylation sites at the very *N* or *C* terminus of proteins but design a peptide library which better reflects situations in loops and intrinsically disordered regions in protein domains, we decided to attach four amino acids *N* and *C*-terminal of pT/pS, plus a *C*-terminal Lys for an additional charge in MS analysis. This was required to keep the diversity and, at the same time, the synthesis quality high. Considering the characteristic of being the smallest chiral amino acid, Ala is routinely used in alanine scanning libraries (i.e. point mutations to Ala) or inverse alanine scanning libraries (i.e. all AA are Ala, and single residues are mutated to other amino acids, PMID 11479122; PMID 10908667). Inversed alanine scans have also been applied to Tyr phosphatases with small sets of peptides (PMID 21719704). We have explained this now better in the manuscript (lines 122-127).

Of note, Ala in fixed positions is completely excluded from our biological analysis. But in our setup Ala is not only used for fixed positions, but is also included for random incorporation in mutated positions. Therefore we also study the effect of Ala as amino acid determinant and control for the reviewer's concerns in an unbiased manner. Our analysis of Ala in randomized positions actually demonstrates that Ala can be considered a quite neutral placeholder, both for PP1 and PP2A (Fig. 2b).

6. Fig. 2B: The patterns for PP1 and PP2A look pretty similar. Which amino acid preferences are statistically significantly different between PP1 and PP2A?

Authors: We have now added an additional heat-map comparing the median-normalized changes between PP1c and PP2Ac and analyzed the differences using the Fisher's exact test (Fig. 2b). Amino acids that behave significantly different between PP1c and PP2Ac with

an adjusted p-value <0.01 are highlighted in bold in the heat-map comparing PP1c/PP2Ac. We would like to thank the reviewer for this suggestion, since the statistical significances obtained even for very small differences between PP1 and PP2A (see Source Data) demonstrate the power of the amino acid coverage in MS obtained from equimolar incorporation in this library-based approach.

Furthermore, this new comparison also brought to our attention an apparently very strong differential effect of Pro in +1. We have therefore synthesized the *Cterm* peptide AAAApTPFGAK. Like in the case of basic and acidic preferences (Fig. 2e), our predictions based on the heat-maps could be recapitulated in non-competitive phosphatase activity assays (Supplementary Fig. 3c). We find that PP2A is unable to dephosphorylate the pTP-peptide, whereas it is a good substrate for PP1. This effect was significantly less pronounced but still observed with statistical significance on the protein level (see Fig. 4). While at first sight this may appear to speak against our methodology, this observation actually is perfectly in line with an early finding which demonstrated that in order to dephosphorylate pSer/Thr with Pro adjacent at +1 position, PP2A requires stabilization of Pro in the *trans* conformation by the isomerase Pin-1 (PMID 11090625), which is not present in the library approach.

7. Fig. 3A):

-Doesn't CalA in the lysate inhibit also the recombinant PP1c and PP2Ac?

Authors: Calyculin A is a highly potent inhibitor of PPPs and is used in the low nM range for efficient inhibitions of endogenous PP1/PP2A within minutes in mammalian cells. We have titrated these ratios multiple times in our laboratory and have found that by adding 20nM Calyculin A to cell culture medium before lysis and to the lysis buffer, we can stabilize the phosphorylation of well-defined PP1/PP2A substrate sites (such as Ser259 of CRAF/Raf-1, PMID 11494123; PMID 12932319; PMID 10801873; PMID 30338897). Please see below for the data, which was published by us in PMID 30338897. We cite this paper now in the text to refer to this experiment. Recombinant phosphatase was added after Calyculin A treatment of cells and lysate to a final concentration of 1 μ M. Therefore, in our setup Calyculin A preferentially inhibits endogenous PP1/PP2A and nM concentrations of Calyculin A can not outcompete μ M phosphatase. To make this more apparent we added the used concentrations in the text.

Fig. Legend: Cells were incubated with 20 nM Calyculin A and lysed in lysis buffer containing 20 nM Calyculin A. To the samples on the left, PP1c was added to a final concentration of 1 μ M (see also PMID 30338897).

- What is the approximate molar ratio between recombinant PPase and substrate proteins in the lysate?

Authors: Concerning the molar ratio of recombinant PPase and substrate proteins we can only present to the reviewer an estimate, which we also calculated for the design of our experiments. For exact numbers, the unknown dynamic range and abundance of the PP1/PP2A substrates would need to enter the equation. However, we can provide a conservative approximation based on the deep proteome of HeLa cells (PMID 22068331; PMID 22278370; PMID 28601559):

- We know that in the assay presented in Fig. 3a, 1mg total HeLa cell lysate are subjected to dephosphorylation by 1 μ M phosphatase in a total volume of 200 μ L.
- It is an established fact that PP1 and PP2A holoenzymes dephosphorylate several hundred substrate proteins covering the full spectrum of functions, such

as membrane proteins, histones, cytoskeletal proteins and transcription and translational fractions (PMID 22284538).

- It has been shown by Nagaraj et al. and others, that the aforementioned types of proteins are among the most abundant fraction of the total proteome, but for a conservative estimation, we will assume an equal protein distribution.
- Therefore, in this conservative calculation, at least 10,000 proteins in the human proteome / 100 substrates = 1% of the 1mg total protein would constitute PP1/PP2A substrate proteins.
- The average molecular weight (MW) of the HeLa proteome was identified to be in the range of 25-50kDa. For our conservative calculation we will take 100 kDa.
- In such a conservative approximation, 1% of 1mg protein = 10µg substrate proteins with an average MW of 100 kDa would result in a molar ratio of 500 µM substrate protein to 1 µM phosphatase, but given the fact that ultradeep phosphoproteomics show that multiple substrate sites are found on one protein (PMID 25159151), it is more likely to be in a range of <<1:500, making dephosphorylation reactions in our setup a quite selective event comparable to *in vitro* kinetic measurement conditions.

8. In. 240: This is a very important potential caveat of the study. Please demonstrate the degree of incorporation of recombinant PP1c and PP2Ac to holoenzyme complexes by gel-filtration analysis and affinity purification coupled with MS.

Authors: Since the goal of our study was to test intrinsic amino acid determinants of the catalytic subunits independent of regulatory subunits this is indeed an important control experiment. Therefore we have developed a setup enabling us to analyze the very same assay presented for phosphoproteomics (Fig. 3a) by gel-filtration. We only introduced His-tags for recombinant phosphatase to be able to discriminate between recombinant and endogenous phosphatase using an α -His antibody in case holoenzymes would be observed. Since the use of detergents in the lysis of HeLa cells interferes with detection of UV-absorbance, we first ensured by Coomassie staining that the use of the detergent does not alter elution profiles of a gel filtration standard ranging from 30-700kDa (Supplementary Figure 5). We then again subjected 1 mg HeLa cell lysate to dephosphorylation with recombinant PP1c or PP2Ac as presented in Fig. 3a. Subsequent gel filtration showed no shifts towards higher molecular weights compared to injecting recombinant phosphatase alone (Supplementary Figure 6). As a positive control we incubated PP1c with the recombinantly purified regulator Inhibitor-2 (I-2/IPP2/PPP1R2, Supplementary Figure 7b) demonstrating that our assay setup would preserve the complex formation of potential holoenzymes. To give the reader an impression of the size-shift that would be expected from PP1/PP2A holoenzyme formation, we also extracted manually curated sets of well-defined PP1/PP2A regulatory subunits based on the HUGO gene nomenclature database and analyzed their average MW. This analysis is shown in Supplementary Figure 7a. With a median MW of 84 (PP1) and 57 (PP2A) kDa one would expect a clear shift towards complexes >100kDa. This possibility can now be excluded given the resolution and results of our gel filtration experiments. Since the points of the reviewer could already be addressed at this stage and there were no higher MW complexes, in which a potential regulatory protein would need to be identified by MS, we think that the presented analysis by Western blotting answers the reviewer's important point.

9. 3D) Are these enrichments statistically significant?

Authors: We have now added a Fisher's Exact Test comparing sites with a fold-change of 0>-1 to sites with a fold-change <-1 and incorporated the results in Fig. 3d. Since the basis for these figures are datasets with >500 p-sites per category, we obtain high statistical significance with an adjusted of p-value <2.5x10⁻⁶.

10. 3E) Similar to 2B, which amino acid preferences are statistically significantly different between PP1 and PP2A? Even more importantly, after calculating the significances for both 2B and 3E, what are the amino acid statistically significant amino acid preferences that are consistent between both methods (Fig. 2 and 3). This would be very important and clear take a home message from the paper with statistical power.

Authors: As for Fig. 2b, we have now also added a heat-map highlighting the differences between PP1c and PP2Ac as Fig. 4b. Again, using Fisher's exact test, we find that Lys in Pos. -1,-4,-5 and +2, Pro in +1 and Arg in -1 and -3 are significantly different between PP1c and PP2Ac. This is in agreement with the current lines of argumentation of the manuscript. Importantly, all of these residues except Lys in -5, which was not tested in the library setup of -4 to +4, also were identified to be preferred by PP1c and PP2Ac in the library setup with statistical significance. This underlines the importance of our observations and we have now incorporated this clear take-home message in the main manuscript (lines 246-250) and it is also depicted in the newly added Fig. 8d.

11. Fig. 4: It is unclear whether this analysis is based on data from Fig. 2 or Fig. 3? Again, related to comments above, similar analysis should be done with both datasets.

Authors: Fig. 4 is now Fig. 5. We have now made a clear statement in the main text (line 261), as well as in the figure legend, that all data shown in Fig. 5 is based on data on the protein level acquired by phosphoproteomics. Because the data largely agree between the two methods, and because in the library we work with 14 different amino acids, we did not do this analysis with the library data. We have carried out a fisher's exact test for Arg in Pos. -3, which is in the focus of the subsequent analysis. Arg in PP1c-sensitive p-sites compared to PP2Ac sensitive p-sites is significantly enriched with a p-value of 0.019 and we have incorporated this analysis in the manuscript.

12. Ln. 276. Is the authors final conclusion that after all there is no aminoacid preference for PP2Ac?? This is very confusing as related to overall message of the paper. Please clarify.

Authors: We appreciate that the reviewer caught this unintended confusion. PP2A has amino acid preferences around the pS/T, as can be seen in figures 2 and 4 of the current manuscript. What we noticed in the motif analysis of Fig. 5a is that these preferences do not translate into a known sequence motif for other proteins such as kinases. For PP1, the Arg-preference relates to the RxxpS/T motif of kinases and 14-3-3 proteins. This is the difference here. We have changed the text accordingly (lines 261-263).

13. Ln. 281-283: The alternative explanation would be that as PP1 functions as a dimer it more easily assembles to holoenzyme than PP2A in these in vitro lysate conditions and therefore it shows more strict substrate selectivity. This further indicates that authors need to provide the experimental evidence requested in my comment 8.

Authors: As requested by the reviewer, the control gel-filtration experiments presented in the new Supplementary Figures 5 and 6 now also control for this possibility. No holoenzyme and no dimer formation were observed in the conditions of our phosphoproteomics assays. In addition, in the context of other projects we have already looked into dimer formation of PP1c (for example for PMID 30403291), and we only observed very small amounts of dimer upon concentrating recombinant PP1c to concentrations in the high mM range, which provides further support that at the nM- μ M concentrations used in this manuscript, dimer formation does not influence substrate selectivity.

14. Fig. 4C; Ln. 286-291: Are these enrichments statistically significant between PP1 and PP2A.

Authors: As the reviewer suggests, these sets of p-sites (now presented in Table 1 to adhere to the journals' formatting guidelines) are found to be exclusively dephosphorylated by PP1c or PP2Ac. The starting point of this filtering is based on Fig. 5b of the revised manuscript, which are p-sites that are different between PP1 and PP2A according to a t-test (unpaired, p-value >0.01). In the revised manuscript we have highlighted this fact better in both, the main text (lines 275-276) and the figure and table legends.

15. Ln. 38: I would prefer that the authors would not strengthen the misconception of the imbalance of kinase and phosphatases. Human genome has 40 genes that code for the

catalytic subunit of S/T PPases, but at least as many functional PPases (as functional complexes) as there are kinases.

Authors: We agree that it is a widespread misconception that the smaller number of catalytic phosphatase genes compared to kinases implies reduced specificity. We have stated this in several previous publications as well. Since we agree with the reviewers' view that at the level of holoenzymes, phosphatases reach a complexity comparable to kinases, we paid attention to communicating this fact better in the revised manuscript (lines 39-44).

16. Ln. 180-181: "that the close proximity of Pro to the pSer/Thr could have suggested an active-site-mediated recognition of the SP motif" For a non-expert reader this is hard to understand

Authors: We thank the reviewer for pointing this out, and have rephrased this (lines 171-172).

17. Ln. 259: "class I sites (localization probability of phosphorylation >0.75 " For a non-expert reader this is hard to understand

Authors: Also here, we have adapted the wording accordingly (line 201).

Reviewer #2 (Remarks to the Author):

The authors in the manuscript entitled “Dissecting the basic substrate specificity of phosphoprotein phosphatases 1 and 2A through phosphoproteomic approaches” have explored large phosphopeptide library to show the pThr specificity of PP1 and PP2 phosphatases. The generation and validation of a reliable phosphopeptide library is crucial for this study. Therefore, authors have purified and verified great deal of different phosphopeptides to carry out PP1 and PP2A dephosphorylation assays. The biggest flaw effecting the whole study is the fact that only recombinant catalytic subunits (PPP1CA & PPP2CA) were used in the dephosphorylation assays. PP1 and PP2A, as well as other protein phosphatases form large protein complexes with multiple regulatory subunits regulating their activities (for example PMID: 27880917 and PMID: 28330616). The main open question is how and if the substrate specificity/preference changes when the PP1 and PP2 regulatory subunit(s) are also used in the assays?

Nevertheless, the authors also tested their conclusion by *in vivo* (HeLa cell line) phosphoproteomics analysis, where both PP1 and PP2 again showed intrinsic preference for pThr sites. They have also investigated if this preference is somehow affected by the any presence of other amino acids around the PThr/pSer. In this quest they have found RXXpS motif, in several PP1 and PP2 substrates as well as in 14-3-3 binding site. One of the identified substrates is GAB2 which also bind with 14-3-3 binding site containing proteins (14-3-3 beta) have been dephosphorylated by PP1 *in vitro*, leading to disruption of GAB2-14-3-3 interaction. This extent of dephosphorylation was also monitored by MS using label-free quantification, where they found PP1 specificity towards RXXpS motif.

Authors: We would like to thank the reviewer for the feedback on our manuscript. We address his/her main open question below.

Major criticism:

The authors should verify how the presence of the regulatory subunit(s) affect PP1 and PP2A substrate specificity.

Authors: This criticism addresses the main open question of the reviewer “how and if the substrate specificity/preference changes when the PP1 and PP2 regulatory subunit(s) are also used in the assays?” as well as “The biggest flaw effecting the whole study is the fact that only recombinant catalytic subunits (PPP1CA & PPP2CA) were used in the dephosphorylation assays”.

It appears that we have not been clear enough on the purpose and design of this study, which also reviewer 1 remarked, and we apologize for that. Importantly, we do not argue at any point against established models of substrate recruitment by regulatory subunits. The question rather was whether the catalytic subunits contribute with sequence preferences around the p-site to the holoenzyme specificity and dephosphorylation kinetics. Therefore, we had to use the catalytic subunits in our study; it was not a flaw.

The motif preference of PP1 and PP2A on the holoenzyme level has been studied in several phosphoproteomic and computational approaches (Refs. 8, 24-27). But it was unclear whether parts of this biologically relevant, holoenzyme preference are coming from the catalytic core protein. The major criticism/question of the reviewer therefore is basically answered by what was already known about the substrate specificity on the holoenzyme level and was the basis of the study, not the question of the study.

Nevertheless, in order to show how our findings relate to holoenzymes in detail, we analyzed eight structural papers (refs. 11-13, 48-52) and one computational paper (ref. 8) regarding the influence of holoenzymes on the preference of the PP1 catalytic subunit for basophilic motifs and on the ability of PP2A to recognize them as holoenzyme while not having an intrinsic preference at the catalytic subunit level (contrary to what the reviewer understood “they have found RXXpS motif, in several PP1 and PP2 substrates”). Furthermore, for PP1 this preference led us to test GAB2 as a new substrate in cells (Fig. 6). We additionally analyzed four cellular functional studies (refs. 24-27) on holoenzymes to corroborate that the pThr preference indeed comes from the subunits. Beyond this, holoenzymes are diverse and have to assemble correctly with the correct, sometimes unknown components, and often the expression of the full-length regulatory proteins cannot be accomplished. Their use in *in vitro* experiments has led to contradictory results (e.g. PMID 28759048; PMID 29618508).

Therefore, functional studies of holoenzymes are best carried out in cells, as seen in refs. 24-27. Picking a few from the over 350 possible ones would not yield a representative, meaningful outcome. For all these reasons, the use of a recombinant holoenzyme in our study would not add any further reliable, representative information, particularly considering the availability of the many structural and cellular studies that we analyzed and that were the basis of this study.

We again thank the reviewer for pointing out this unclear part of our manuscript, which we have now addressed thoroughly.

Minor comments:

Only one cell line was used in vivo assay. The use of (an)other cell line(s) would strengthen the manuscript. Additionally for authentication of the cell line statement “to the best of our knowledge these are HeLa cells” is not sufficient. See for example a great study on the HeLa heterogeneity by the Aebersold group “PMID: 30778230”

Authors: In the revised manuscript we now present our efforts to extend the findings in the in vivo live-cell microscopy to other cell lines. We were able to apply the peptide tool PDP(m)-*NaI* in two additional cell lines derived from epithelial tissue, namely Caco2 and SW480 (presented in Supplementary Figure 8 and Supplementary Movies 4-9). Again, we made use of the image analysis pipeline presented in Fig. 6d and were able to demonstrate that the selective GAB2-membrane recruitment upon PDP-*NaI* treatment (with p-values >0.001 according to a Wilcoxon-test) happens also in cell lines derived from colon cancer. We would like to thank the reviewer for this suggestion, since this newly acquired data now extends the scope of our findings from cervix cancer to additional types of cancer in colon.

We also fully agree with the Reviewer's views that cell line authentication and integrity is an essential factor towards research reproducibility. To support these important efforts, we subjected HeLa Kyoto cells, as well as the newly introduced Caco2 and SW480 cell lines to authentication by an external, independent service provider (Labor f. DNA-Analytik, Freiburg, Germany) and mycoplasma tests (Eurofins/GATC, Ebersberg, Germany). The obtained Pm-values (likelihood of random match) were 4.26×10^{-15} (HeLa Kyoto), 5.86×10^{-11} (Caco2) and 3.84×10^{-12} (SW480), respectively. All cell lines were found to be free of mycoplasma. We will also provide the underlying certificates in our Source Data.

Minor corrections

-Page 19 , line 10 ----(Fig 6a) should be Fig 5c,
line 11-----Fig 6b should be Fig 5d,
line 12----- (Fig 6c) should be Fig 5e,
line 14----- (Fig 5b) should be Fig 5c.

Authors: Thank you for highlighting these errors. In the resubmission we have corrected the labeling accordingly.

-Figure 5a should show error bars. In its legends, e, should be legend d and f, should be legend e.

Authors: We thank the reviewer for pointing this out. For Fig. 5a of the initial submission we have now made successful efforts to optimize the Western Blot setup in order to increase chemiluminescent signals. We can now provide four independent experiments of the GAB2-dephosphorylation assay all of which can be quantified, yielding reproducible results with statistical significance (Fig. 6a,b). We have also adapted the quantification towards the Guidelines of Nature Communications to display single data points. In addition, we corrected the mislabeled figure legends.

REVIEWERS' COMMENTS:

Reviewer #1 (Remarks to the Author):

Authors have done excellent work in responding to my comments and criticism. There are only minor issues remain gin to be addressed (numbering refers to numbering of the original comments):

1. The schematic figures in 8d,e are very helpful. Meanwhile two important PP2A phosphoproteome studies have been published and for the readers sake it would be useful to shortly comment how the results of those studies fit to the authors model (PMID: 32400009, PMID: 32071079). Even though this is not a review article, again to help the reader, all such studies cited in the paper could be referenced together in the introduction after a sentence like this "Several recent studies have identified substrates for PP1 and PP2A by different approaches....."

2. It is still not crystal clear from the title that the study focuses on catalytic subunits, but not on holoenzymes. Why not to use the protein names of the catalytic subunits PP1C and PP2AC in the title instead of PP1 and PP2A?

8. & 13. Authors effort to respond to these comments by gel filtration analyses is acknowledged. However, there is some evidence for shift towards higher molecular weight complexes with recombinant PP1c and this should be clearly mentioned in the text.

12. The text in the lines 261-263 is still not quite clear. Why not to write out that you mean e.g. kinase target motifs with "motif" as it is explained in the response to me. "...a known motif, such as kinase target motif..."

-The manuscript is so full of details and complicated data/interpretations that it would be a great favor for the reader if the entire manuscript would be proof-read and edited by a professional scientific editing service for maximum clarity

- Next time, please label the figures with figure numbers

Reviewer #1 (Remarks to the Author):

Authors have done excellent work in responding to my comments and criticism. There are only minor issues remain to be addressed (numbering refers to numbering of the original comments):

1. The schematic figures in 8d,e are very helpful. Meanwhile two important PP2A phosphoproteome studies have been published and for the readers sake it would be useful to shortly comment how the results of those studies fit to the authors model (PMID: 32400009, PMID: 32071079). Even though this is not a review article, again to help the reader, all such studies cited in the paper could be referenced together in the introduction after a sentence like this "Several recent studies have identified substrates for PP1 and PP2A by different approaches....."

We thank the Reviewer for her/his feedback and the acknowledgement of our efforts to improve the manuscript. We are aware of both publications and they are nicely complementary to our work, particularly the substrate specificity study PMID 32400009. We have now added them as references 53 and 55, and mention them in lines 505-506, 512 and 517-520 (line numbering with track changes).

2. It is still not crystal clear from the title that the study focuses on catalytic subunits, but not on holoenzymes. Why not to use the protein names of the catalytic subunits PP1C and PP2AC in the title instead of PP1 and PP2A?

We agree with the Reviewer and changed the title following the Editor's suggestion to "Dissecting the sequence determinants for dephosphorylation by the catalytic subunits of phosphatases PP1 and PP2A". To further improve this point, we have also made efforts to adapt also all labels in figures to mention consistently PP1c and PP2Ac instead of PP1 and PP2A.

8. & 13. Authors effort to respond to these comments by gel filtration analyses is acknowledged. However, there is some evidence for shift towards higher molecular weight complexes with recombinant PP1c and this should be clearly mentioned in the text.

Indeed, western blots show that a very small fraction of recombinant phosphatase might shift towards higher molecular weight complexes and we have now changed the wording in lines 292-300 (incl. track changes) accordingly.

12. The text in the lines 261-263 is still not quite clear. Why not to write out that you mean e.g. kinase target motifs with "motif" as it is explained in the response to me. "...a known motif, such as kinase target motif..."

We thank the Reviewer for pointing this out and we have now explicitly mentioned kinase motifs as an example in line 319 (incl. track changes).

-The manuscript is so full of details and complicated data/interpretations that it would be a great favor for the reader if the entire manuscript would be proof-read and edited by a professional scientific editing service for maximum clarity.

We agree with the Reviewer that the manuscript is complex due to the interdisciplinarity of methodological approaches and aspects of MS analysis as well as the biological aspects presented in the manuscript. For increasing clarity, we have therefore, over several rounds, already exchanged the manuscript with native

speakers and scientists not familiar with the project. We have also worked together with a professional scientific writer (Christina Gross, please see the acknowledgements). In the current revision, we have tried to improve the clarity further by using percentages instead of solely the total numbers to describe the results in a relative, comparative manner.

- Next time, please label the figures with figure numbers

We are sorry for this inconvenience. Upon initial submission figure numbers were depicted in the actual graphic, but we removed them for resubmission in order to fulfill the journal's requirements for graphical formatting.